



# ACROPOLIS: Munich Urban $CO_2$ Sensor Network

Patrick Aigner[1], Jia Chen[1], Felix Böhm[1], Mali Chariot[2], Lukas Emmenegger[3], Lars Frölich[1], Stuart Grange[3,4], Daniel Kühbacher[1], Klaus Kürzinger[1], Olivier Laurent[3], Moritz Makowski[1], Pascal Rubli[3], Adrian Schmitt[1], and Adrian Wenzel[1]

[1]Technical University of Munich (TUM), Munich, Germany
[2]Laboratoire des Science du Climat et de l'Environnement (LSCE/IPSL), Gif-sur-Yvette, France
[3]Swiss Federal Laboratories for Materials Science and Technology (Empa), Dübendorf, Switzerland
[4]University of Bern, Bern, Switzerland

**Correspondence:** Patrick Aigner (patrick.aigner@tum.de) and Jia Chen (jia.chen@tum.de)

**Abstract.**

Urban areas are major contributors to anthropogenic $CO_2$ emissions, yet detailed monitoring remains a challenge due to the cost and operational constraints of traditional sensor networks. As a scalable alternative, we established the ACROPOLIS (Autonomous and Calibrated Rooftop Observatory for MetroPOLItan Sensing) network in the Munich metropolitan area, using

mid-cost sensors to enable dense, city-scale observation. This work outlines the development of the hardware and software of the system, its performance and the first year of operation, during which more than 70 million $CO_2$ measurements were collected in urban, suburban and rural environments.

The primary goal was to evaluate whether mid-cost Vaisala GMP343 sensors, when combined with manufacturer internal corrections and environmental stabilization, can reliably measure $CO_2$ concentrations with sufficient accuracy to resolve urban

gradients. We implemented a fully automated 2-point calibration procedure using synthetic dry reference gases and conducted a multi-week side-by-side comparison with a high-precision Picarro reference instrument to assess sensor performance.

Our results show that, despite inter-sensor variability in temperature sensitivity, the hourly aggregated mean root mean square error (RMSE) of all sensors is 1.16 ppm with a range of 0.57 to 2.58 ppm. For the specific sensor housed in our second-generation enclosure with PID-controlled heating, the performance improved from 0.9 to 0.6 ppm RMSE. Analysis of spatial

and temporal patterns reveal distinct seasonal cycles, urban–rural concentration gradients, and nighttime accumulation events, consistent with expected biogenic and anthropogenic activity, and atmospheric transport mechanisms.

We conclude that mid-cost urban networks can provide scientifically valuable, spatially highly resolved greenhouse gas observations when supported by appropriate calibration and stabilization techniques. The open-source design and demonstrated performance of the ACROPOLIS network establish a blueprint for future deployments in other cities seeking to advance

emissions monitoring and urban climate policy.



# 1 Introduction

Global atmospheric observations consistently show that concentrations of well-mixed greenhouse gases have been steadily increasing year by year, with the earliest continuous measurements dating back to 1958 at the Mauna Loa Observatory in Hawaii (Keeling et al., 1976). Among these, carbon dioxide ($CO_2$) is the largest contributor to anthropogenic climate change, accounting for the biggest share of both total volume and radiative forcing effects (Szopa et al., 2021).

Recent assessments by the Intergovernmental Panel on Climate Change (IPCC) confirm that global net emissions are increasing, although the annual growth rate has slowed slightly in the past decade. In particular, emissions in all major sectors, including energy, industry, transportation, buildings and agriculture / land use, have increased between 2010 and 2019 (Dhakal et al., 2022).

Cities represent hotspots for human activity and energy consumption, and global urbanization is rapidly accelerating, with urban population projected to increase from over 55 % (approximately 4.2 billion people) today to approximately 67 to 70 % by 2050 (Dodman et al., 2022). Although this change will significantly increase total net emissions within urban areas, the trends in per capita emissions are less certain. Empirical studies show that dense, transit-oriented cities generally exhibit lower per capita $CO_2$ emissions compared to sprawling, car-dependent urban areas (Kennedy et al. (2009), Hoornweg et al. (2011), Jones and Kammen (2014)). However, cities are not homogeneous. They encompass diverse neighborhoods shaped by factors such as historical development, transportation infrastructure, energy supply, and industrial activities. Consequently, city-wide averages for per capita emissions can mask considerable intra-city variations, with dense urban cores typically emitting significantly less $CO_2$ per capita than surrounding suburban areas (Hoornweg et al., 2011). To accurately characterize this intra-city heterogeneity and inform targeted mitigation strategies, it is essential to understand long-term emission trends at high spatial resolution in urban areas.

Traditional urban greenhouse gas (GHG) monitoring networks (Bréon et al. (2015), Turnbull et al. (2015), Davis et al. (2017), Verhulst et al. (2017), Mueller et al. (2018), Mitchell et al. (2018), Bares et al. (2019), Karion et al. (2020)) typically rely on a limited number of high-precision instruments, such as Cavity Ring-Down Spectroscopy (CRDS) analyzers, to capture regional $CO_2$ signals. Although these instruments provide excellent accuracy (sub-ppm), their high cost and operational requirements constrain the total number of sites, resulting in insufficient coverage to resolve fine-scale $CO_2$ variations at the neighborhood scale, which according to Turner et al. (2016) is approximately 1 to 2 km.

Although increasing the number of observation sites is critical, it introduces a trade-off between budget and accuracy. Atmospheric inversion modeling studies (Turner et al. (2016), Wu et al. (2016)) highlight that a sensor accuracy of approximately 1 ppm (1 $\sigma$) is needed to effectively resolve urban line and point sources.

Mid-cost nondispersive infrared (NDIR) sensors, including models such as Vaisala GMP343, LI-COR LI-850, and SenseairHPP (prototype), offer a promising solution for dense urban networks, but require careful calibration and stabilization to consistently achieve the targeted accuracy (Arzoumanian et al. (2019), Delaria et al. (2021), Cai et al. (2024), Lian et al. (2024), Grange et al. (2025)). The primary challenge with using NDIR sensors is that their laboratory accuracy often degrades under field conditions due to fluctuations in temperature, pressure, and humidity. This work specifically focuses on the Vaisala GMP343



sensor, which offers a manufacturer-reported accuracy of up to 1 ppm under controlled laboratory conditions and is at least one order of magnitude cheaper than high-precision CRDS instruments. Although pressure-induced variations are relatively uniform across GMP343 sensor units (Shusterman et al., 2016), GMP343 sensors demonstrate unique sensor-specific temperature sensibility that requires individual corrections (Delaria et al., 2021). In contrast to other available NDIR sensors, Grange et al.
(2025) shows that the GMP343 probe already internally corrects a pressure-broadening effect of $H_2O$ molecules, confirming that probe has a reliable factory-based correction for water-induced effects.

In addition to their sensitivity to environmental conditions, NDIR sensors drift over time, necessitating correction strategies to maintain performance. Strategies consist of periodic calibrations with one or more calibration cylinders at varying intervals (Arzoumanian et al., 2019; Park et al., 2021; Lian et al., 2024; Grange et al., 2025), comparisons with nearby high-precision
reference instruments during uniform background conditions (Shusterman et al., 2016; Müller et al., 2020), or the use of machine learning techniques (Martin et al., 2017).

One key objective of the ICOS Cities project (2021–2025) is to integrate mid-cost sensors into rooftop deployable enclosed systems and to establish prototype networks in the three pilot cities: Paris (Lian et al., 2024), Munich, and Zurich (Grange et al., 2025). Although the cities collaborated closely, each adopted a different system configuration, including choice of sensor
type, indoor or outdoor deployment, calibration methods, and data processing strategies. All systems aimed at a cost of 5–10 k€ per unit and an accuracy of 1 ppm RMSE.

This study presents the development, deployment, and operation of Munich's mid-cost sensor network ACROPOLIS (Autonomous and Calibrated Rooftop Observatory for MetroPOLItan Sensing), consisting of 20 systems deployed across 17 sites. The network uses off-the-shelf Vaisala GMP343 sensors without any laboratory characterization, relying instead on manufac-
turer's correction functions and environmental stabilization. By reducing pre-deployment effort and leveraging the GMP343's built-in corrections, the network achieves the target accuracy of 1 ppm RMSE while significantly reducing both the cost and the time required to scale the network. All stations are equipped with IoT connectivity (4G/NB-IoT via MQTT), enabling remote software updates, configuration adjustments, and real-time data transmission. Routine operations are fully automated, including a 2-point calibration procedure that mitigates sensor drift and ensures long-term measurement stability. As a result,
maintenance is largely limited to hardware failures and periodic calibration gas replacement, with the goal of reducing on-site visits to once per year. Sensor performance is validated through side-by-side field comparisons with a high-precision reference instrument. We show whether this mid-cost network can reliably resolve neighborhood-scale $CO_2$ variability and serve as a scalable blueprint for future urban greenhouse gas monitoring systems.

## 2  Methods

This section describes the technical implementation of the ACROPOLIS network. We begin by describing the hardware architecture and selected components of the field-deployable sensor systems. We then present the open-source software stack, which includes system automation, data transmission, and remote device management. The following subsections detail the processing steps applied to raw sensor output, encompassing on-device corrections and post-processing procedures. Finally, we



describe the quality control routines implemented and provide an overview of the spatial deployment and operational aspects
of the sensor network.

## 2.1 Hardware

To enable autonomous operation, the Vaisala GMP343 probe must be integrated into a reliable field-ready system. We decided
to carry out the development and assembly in-house to enable flexible prototyping and iterative design improvements.

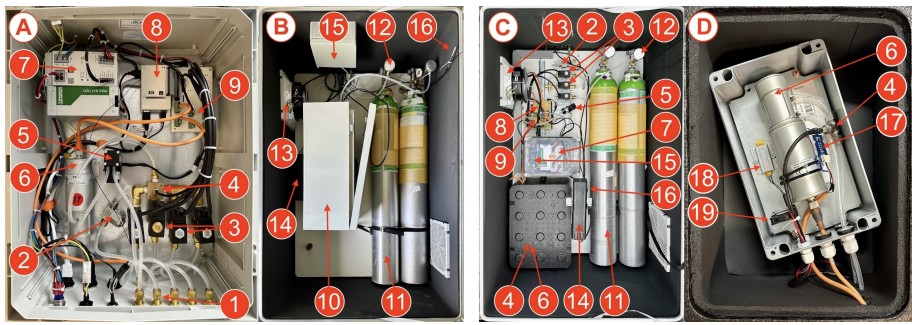

**Figure 1.** Overview of the ACROPOLIS hardware architecture, illustrating the first-generation system (A + B) and the second-generation
system (C + D). Key components are highlighted using overlapping numbered labels: (1) 4 Inlets and Outlet, (2) 2 µm particulate filter, (3)
2/2 solenoid valves (1 always open, 3 always closed), (4) Brass manifold with in-flow sensor breakout board (BME280, SHT45), (5) eccentric
diaphragm pump, (6) GMP343 Vaisala probe, (7) UPS with AGM battery, (8) Raspberry Pi 4 with Waveshare 4G module, (9) Mainboard with
BME280 sensor, (10) measurement unit, (11) 10 L aluminum calibration cylinders, (12) demand flow regulators, (13) control cabinet fan,
(14) 350 W control cabinet heater, (15) power distribution and Arduino Nano, (16) enclosure temperature sensor, (17) heat box temperature
sensor, (18) 10 W PTC heater, (19) 24 V axial fan

### 2.1.1 Measurement Unit

The measurement unit (Fig. 1, A) houses all components needed for the operation of the GMP343 sensor. The main air
intake is filtered by a 2 µm particulate filter (Fig. 1, A2) to prevent contamination. For internal compensation, the GMP343
sensor receives relative humidity readings inflow from a Sensirion SHT45 sensor (Fig. 1, A4) and pressure readings from a
Bosch BME280 sensor (Fig. 1, A4). Although the BME280 also provides relative humidity readings, the SHT45 offers better
accuracy, particularly under very low relative humidity conditions. To enable operational resilience, the system incorporates
an uninterruptible power supply (UPS) and a battery (Fig. 1, A7), which provides one hour backup power during outages.
A Raspberry Pi 4 (Fig. 1, A8) serves as the central controller of the system, managing all sensors and actuators. For remote
data transmission, the system is equipped with a cellular 4G module (Fig. 1, A8). A custom mainboard (Fig. 1, A9) integrates
terminal block sockets, serves as the main power distribution hub for all components, and houses a second BME280 sensor.





### 2.1.2 Outdoor Enclosure

The measurement unit (Fig. 1, A) is integrated in an outdoor enclosure (Fig. 2 A). The enclosure is modified with additional cartridge silicone sealing to protect the components from all local weather conditions. The inner walls are insulated with ArmaFlex to improve thermal stability. For calibration, two 10 L aluminum calibration cylinders (Fig. 1, B11) are integrated. These cylinders are provided by Westfalen Gas. We used a gas mixture consisting of synthetic air (80 % Nitrogen Grade 6.0, 20 % Oxygen Grade 6.0), spiked with $CO_2$ (400 ppm and 520 ppm Carbon-dioxide Grade 3.0), with a precision of about $\pm 2$ % and

stability of 36 months. The exact $CO_2$ concentration is determined using a Picarro instrument calibrated against the WMO $CO_2$ X2019 calibration scale (Hall et al., 2021), thereby transferring the calibration scale to the calibration cylinders. The cylinders are connected to the system through demand flow regulators (Fig. 1, B12), which release gas flow upon application of underpressure by an eccentric diaphragm pump (Fig. 1, A5). The system is designed to have approximately 0.5 L/min of flow during both calibration and ambient measurements. An additional advantage of demand flow regulators is that, while they

are not in use, only the regulator is pressurized, thereby minimizing the risk of leakage from downstream tubing or valves. Four 2/2 solenoid valves (Fig. 1, A3) allow to switch between the main air intake and up to three additional lines. An external LTE antenna mounted on the aluminum outdoor enclosure provides 4G cellular reception. An industrial grade 350 W control cabinet heater (Fig. 1, B14), a control cabinet fan (Fig. 1, B13), and Arduino Nano (Fig. 1, B15) with hysteresis control provide an internal temperature of 25±3 °C. At low ambient temperatures, the heater is activated to maintain the target range, while at

higher temperatures, which are primarily the result of solar-induced heat buildup, the ventilation system prevents overheating.

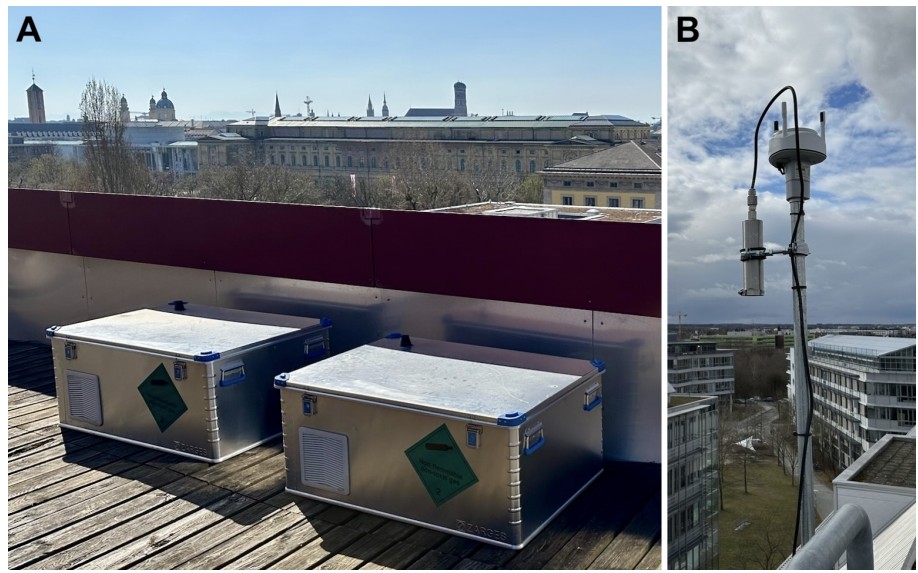

**Figure 2.** (A) Two ACROPOLIS outdoor enclosures deployed on the university rooftop site (TUMR). (B) Air inlet and co-located Vaisala WXT-532 wind sensor mounted on a pole at the GROR site.





### 2.1.3 Inlet and Wind Sensor

To allow flexible placement on structures such as poles in an elevated rooftop position, the system includes an external air intake (Fig. 2 B). It can be connected to the outdoor enclosure (Fig.2 A) with a maximum of 50 m of data cable and air tubing. Accompanying the intake, a Vaisala WXT-532 wind sensor (Fig. 2 B) records wind speed and direction, which are transmitted
together with the $CO_2$ data.

### 2.1.4 Sensor System Generation 2

Based on the experience gained during the first year of deployment (Section 3.3.1), we built a prototype for a second generation system (Fig. 1, C). The main focus was on improving the temperature control and maintainability in the field.

We removed the enclosure of the measurement unit (Fig. 1, B10) and directly mounted individual components on DIN
rails (Fig. 1, C2,3,8,9) to enable quickly changing components in the field. By reducing the number of brass connections and integrating Swagelok tube connectors, we simplified the process of creating an airtight flow through the system.

The main improvement is the addition of a dedicated PID-controlled temperature-stabilized sensor chamber with a temperature sensor (Fig. 1, D17), a 10 W positive temperature coefficient (PTC) heater (Fig. 1, D18), and a small ventilator (Fig. 1, D19). This fully enclosed and small-volume aluminum box maintains the sensor environment at a configurable target temper-
ature ±0.1 °C. A high temperature target eliminates the effects of solar radiation during the hot summer months. The cabinet heater (Fig. 1, C14) and the ventilator (Fig. 1, C13) are unchanged to prevent condensation or freezing, and to ensure a stable environment for the calibration cylinders (Fig. 1, C11). We refer to the updated sensor enclosure as second-generation hardware (also termed v2).

The bill of material of the system for generation 1 is 7800 € , with approximately 50 % attributed to the Vaisala GMP343
sensor and the WXT-532 wind sensor. This excludes consumables, labor, and operational costs. Generation 2 reduced the total system cost to 7300 € .

## 2.2 Software

All software components are fully open-source, ensuring transparency, long-term accessibility, and reproducibility in scientific contexts. Particular emphasis was placed on a modular and reliable architecture that is easy to deploy even for non-expert users.
The system architecture supports flexible customization, is highly scalable, and has low operational overhead by combining containerized deployment, standard communication protocols, and remote update capabilities. All software is implemented as typed Python scripts and is continuously tested via GitHub Actions as part of a Continuous Integration (CI) pipeline.

To minimize technical barriers, the software stack is divided into four components: (1) a middleware component for managing on-device data transmission, remote procedure calls (RPCs) and software updates "Gateway", (2) a component for edge
system automation "controller", (3) a third-party cloud-based data collection platform "ThingsBoard", and (4) a data post-processing pipeline. Detailed documentation and usage instructions are provided in the respective GitHub repositories.



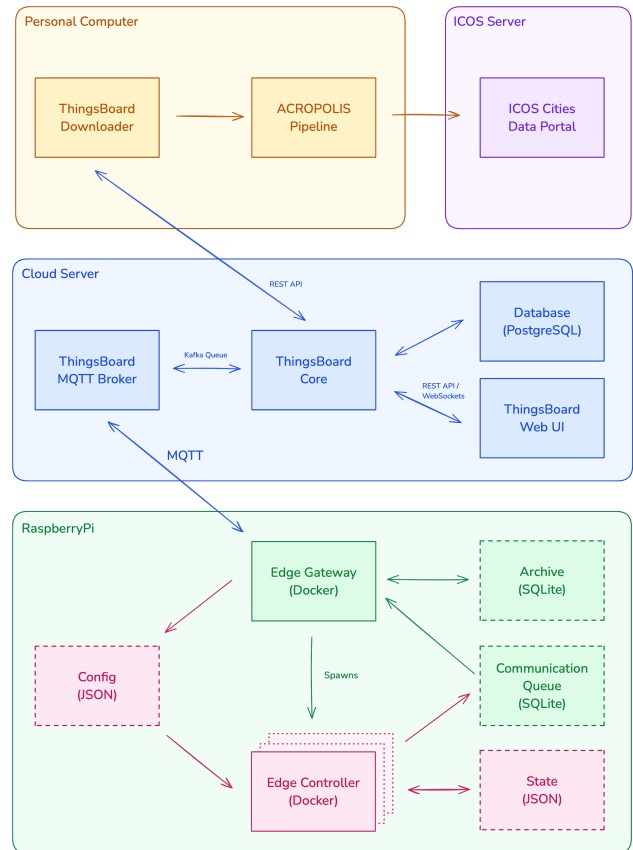

**Figure 3.** Schematic overview of the ACROPOLIS open-source software stack. The diagram illustrates the software components running on different system elements. A Raspberry Pi 4 at each station runs the ACROPOLIS edge software, which controls sensors and actuators and communicates via MQTT with a cloud server hosting a ThingsBoard instance. The ThingsBoard server provides real-time data visualization, device management, and telemetry storage. The ThingsBoard Downloader enables authenticated local retrieval of measurement data, which is processed by the ACROPOLIS post-processing pipeline. Final data products are published to the ICOS Cities data portal for public access.

### 2.2.1 Edge System Automation

Each field station runs Python-based ACROPOLIS edge software (Aigner et al., 2025) on a Raspberry Pi, consisting of a continuously running gateway process and a containerized version controlled controller software managed by the gateway.

The edge gateway is a standalone process that ensures the availability of the station 24/7. It handles MQTT communication with the backend, serves as the endpoint for remote procedure calls, and manages the deployment and versioning of the edge controller container. After successful transmission to the MQTT broker, a copy of the data is archived locally via SQLite and can be re-uploaded in case of transmission failure or data loss. In addition, selected log messages and health checks are also forwarded to Thingsboard for real-time monitoring and diagnostics. The gateway process monitors the software version



assigned to it via ThingsBoard and upon receiving a new software version assignment, fetches the specified version from GitHub and redeploys the controller software using Docker.

The edge controller manages all connected devices through hardware-specific interfaces, organized into reusable and configurable modules. A key feature of the controller is its ability to perform on-device processing for dilution and calibration correction. This allows the system to transmit corrected measurement data in real time, making it immediately available to users and operators through the ThingsBoard dashboard, without the need for additional downstream processing. Measurement data are written to a persistent queue based on a SQLite database that is continuously monitored and processed by the gateway.

### 2.2.2 Cloud-based data collection platform

For cloud-side device management, data collection, and visualization, the project uses ThingsBoard (Thi, 2024), a scalable open-source IoT platform. It acts as the central backend for telemetry exchange, data storage, and real-time visualization through customizable dashboards. ThingsBoard supports remote procedure calls, rule-based alerting, and lifecycle management of connected devices, among other features. Programmatic access is enabled through a REST API. Data transmission between field stations and the backend is secured using TLS.

Measurement data from all stations are transmitted with a frequency of 10 s via MQTT. Internally, Apache Kafka handles asynchronous message queueing and buffering. Measurement data are stored in a PostgreSQL database. ThingsBoard was selected based on its maturity and comprehensive feature set after earlier prototyping with Hermes (Hermes-contributors, 2025) and Tenta (Böhm et al., 2025).

### 2.2.3 Post-Processing pipeline

To support local data validation, correction, and export, the post-processing pipeline is composed of two dedicated tools: the ThingsBoard Downloader (Aigner and Chen, 2025b) and the ACROPOLIS-data-processing (Aigner and Chen, 2025a) module. Both tools support a uniform and reproducible processing workflow, ensuring that all sensor outputs undergo the same calibration corrections, quality checks, and formatting procedures. This standardization allows reproducible comparison between systems.

The ThingsBoard Downloader enables authenticated access to the backend via the ThingsBoard REST API using JWT tokens for authentication. Users can select specific stations and telemetry keys and download compressed measurement datasets for offline analysis. This local copy of the data simplifies data handling and decouples local processing from backend operations.

The ACROPOLIS-data-processing module uses the Polars library for tabular data operations, enabling efficient handling of large datasets. Additional Jupyter Notebooks support calibration tank characterization and performance evaluation with deployed reference instrument. These tools complement the automated pipeline by enabling manual inspection, exploratory analysis, and quality assurance across the entire measurement network.





### 2.3 Data Processing

#### 2.3.1 Sensor Settings

Sensor settings include activated optics heating, linearization, and 10 s averaging of output data. Supplier-provided sensor
internal compensation for temperature, relative humidity, pressure, and oxygen is enabled. Relative humidity and pressure
readings are updated live by external sensors in the gas flow at a sampling frequency of 0.1 Hz. A moving average is used
over 15 consecutive measurements to smooth short-term fluctuations in the low-cost sensor output. During the injection of dry
calibration gas, the humidity of the GMP343 sensor is set to 0 %, while the offset of the SHT45 sensor is determined. The
offset is then corrected for until the next calibration event.

#### 2.3.2 Dilution Correction

The dry air mole fraction of $CO_2$ is calculated in three steps. First, by approximating the saturation water vapor pressure using
the Wagner equation (Eq. 1).

$$p_\sigma = p_c \cdot \exp\left[\frac{T_c}{T} \cdot \left(a_1\theta + a_2\theta^{1.5} + a_3\theta^3 + a_4\theta^{3.5} + a_5\theta^4 + a_6\theta^{7.5}\right)\right] \tag{1}$$

with $\theta = (1 - T/T_c)$, $T_c$ = 647.096 K, $p_c$ = $22.064 \cdot 10^6$ Pa, $a_1$ = -7.85951783, $a_2$ = -1.84408259, $a_3$ = -11.7866497,
$a_4$ = 22.6807411, $a_5$ = -15.9618719, $a_6$ = 1.8012250 (Wagner and Pruß, 2002).

Then the mole fraction of water vapor is calculated (Eq. 2) according to Dalton's law of partial pressures, by dividing the
partial pressure of the water vapor by the total ambient pressure. The ambient pressure ($P_{total}$) is supplied by the BME280
sensor, while the relative humidity (RH) is read from the Sensirion SHT45. Finally, $CO_2^{dry}$ is calculated (Eq. 3) using the
water vapor mole fraction.

$$x_{H_2O} = \frac{P_{H_2O}}{P_{total}} = \frac{p_\sigma \cdot \frac{RH}{100}}{P_{total}} \tag{2}$$

$$CO_2^{dry} = \frac{CO_2^{wet}}{1 - x_{H_2O}} \tag{3}$$

#### 2.3.3 Calibration Correction

Each system is calibrated using an automated 2-point calibration procedure with two synthetic gas calibration bottles filled with
approximately 400 and 520 ppm $CO_2$. The calibration procedure is performed every other day at 03:00 local time using an





alternating bottle sequence. At this time, the ambient temperatures are consistently below the target (25±3 °C for v1, 40±0.1 °C for v2), allowing the control cabinet heater to maintain stable internal conditions. We use the first bottle in the sequence to dry the air channel before the calibration, and alternating cylinders balance the usage of cylinder gas for drying over time. Each calibration bottle is sampled for 10 min, preceded by a 10 min pre-drying phase and a 5-minute flushing period at 2 L/min. The total sequence takes 35 min.

The calibration parameters for slope (m) and intercept (b) are calculated by taking the median of both cylinder runs and are automatically validated after each calibration event to be within ±10 % of the last valid run or are ignored otherwise. The state of all calibrations can be assessed through a ThingsBoard Dashboard for Network Operation, which also offers an option for automated alarms.

$$CO_2{}^{cal} = m \cdot CO_2{}^{dry} + b \tag{4}$$

### 2.3.4 Data Flagging

The collected data is downloaded using the ThingsBoard Downloader and aggregated to 1-minute data and further processed by the post-processing pipeline (Section 2.2.3). Outliers are detected and flagged using a Hampel filter (Pearson et al. (2016), El Yazidi et al. (2018)) configured at a 3-sigma threshold in a sliding window of 120 min. It uses a Median Absolute Deviation (MAD) to detect values that deviate significantly from the surrounding data. The resulting flagged 1-minute dataset is formatted in a CSV format and automatically uploaded to the ICOS Cities Portal as a Level 1 (L1) dataset. Following this, manual operator validation and additional flagging are performed to produce a Level 2 (L2) data set. These are uploaded as one-minute and hourly averaged datasets.

### 2.4 Quality Control

As part of the post-assembly verification, the system is checked for air-tightness to ensure that only gas from the sampling line or calibration tanks is measured. To do this, a vacuum pump is used to apply underpressure, and a manometer is used to monitor the pressure over time. The system passes the quality test if the pressure rise due to leakage remains below 10 mbar/min. The gas flow through the system is checked and tuned using a mass flow meter, and the response of the relative humidity sensors is tested by injecting dry gas.

Redundancy and drift detection were implemented for all in-flow low-cost sensors. By co-locating a BME280 and an SHT45 sensor on the inlet line, we continuously cross-check temperature and humidity readings to identify any long-term sensor drift. Likewise, drift in the inflow pressure sensor is evaluated by comparing its measurements to a second BME280 on the mainboard. This is done by monitoring the pressure difference between the inlet and enclosure over time to detect systematic bias, pump degradation, or potential leaks.

The dilution (Section 2.3.2) and calibration corrections (Section 2.3.3) are available both on device and in postprocessing. This allows cross-validation of live on-device corrections with results from the postprocessing pipeline and enables reprocessing of calibration events.



System performance is evaluated through multi-week side-by-side tests in outdoor conditions, sampling concurrently with a Picarro G2301 reference instrument installed on the rooftop of the TUM campus in the center of Munich. Measurement inlets were placed next to each other for the mid-cost systems and the reference instrument. The location supported up to ten mid-cost systems at a time. During the side-by-side campaign, we gradually deployed systems while replacing open spots with waiting systems. The evaluation periods range from 16 to 85 days, depending on opportunities for deployment. Weather sealing and

outdoor stability were confirmed during this assessment, including heavy rainfall, snowfall, and extreme temperatures.

During the side-by-side and field operation, auxiliary sensor data is continuously sent to the backend for ongoing monitoring and assessment. Automated rule chains in ThingsBoard provide real-time alerts for unexpected behavior of the system and can be customized by the operator.

## 2.5  Network

The ACROPOLIS network is deployed throughout the metropolitan area of Munich. Munich is the third largest city in Germany, located in the southeast, covering an area of 310 km$^2$ and home to approximately 1.5 million people. The prevailing wind directions are from the southwest and east.

The deployment followed a set of criteria designed to ensure representative sampling of a well-mixed urban signal, minimize local contamination, and mitigate risks to buildings and personnel. To reduce the influence of nearby emission sources, we

prioritized (1) rooftops higher than surrounding structures, (2) free-standing buildings, and (3) buildings with district heating or detached central heating systems. (4) Inlets were placed to allow unobstructed airflow, avoiding ventilation outlets and chimneys, especially along the dominant wind directions.

All sites require a reliable power connection to the grid, and the stations are located in secured or restricted access areas to ensure the integrity and operational safety of the system. We relied on pre-existing lightning protection and placed inlets under

its protective coverage. Preference was given to working with building owners who manage multiple properties to streamline deployment logistics and reduce contractual overhead.

### 2.5.1  Network Deployment

Following the deployment strategy, 20 stations were installed across 17 sites in urban, suburban, and rural locations in and around Munich, covering all major districts. The stations were placed indoors (e.g. elevator shafts, technical rooms) or directly

outdoors on the roof. Schools and hospitals offered ideal conditions, as they are typically free-standing, elevated structures distributed throughout the city and are overseen by a centralized authority.

The present deployment includes six school sites (MAIR, PASR, SCHR, BALR, SENR, TAUR), and four hospitals (RDIR, HARR, BOGR, NPLR). Other deployed sites are two city halls (FINR, FELR), private institutions (SWMR, BLUT), and research buildings (DLRR, TUMR, GROR). Although most of the sites rely on district heating or detached block heating, two

locations (MAIR, BALR) have close-by gas heating infrastructure, which can influence the measurement signal under specific conditions, such as low wind speeds. We placed inlets in a way to support uncontaminated measurements from the dominant




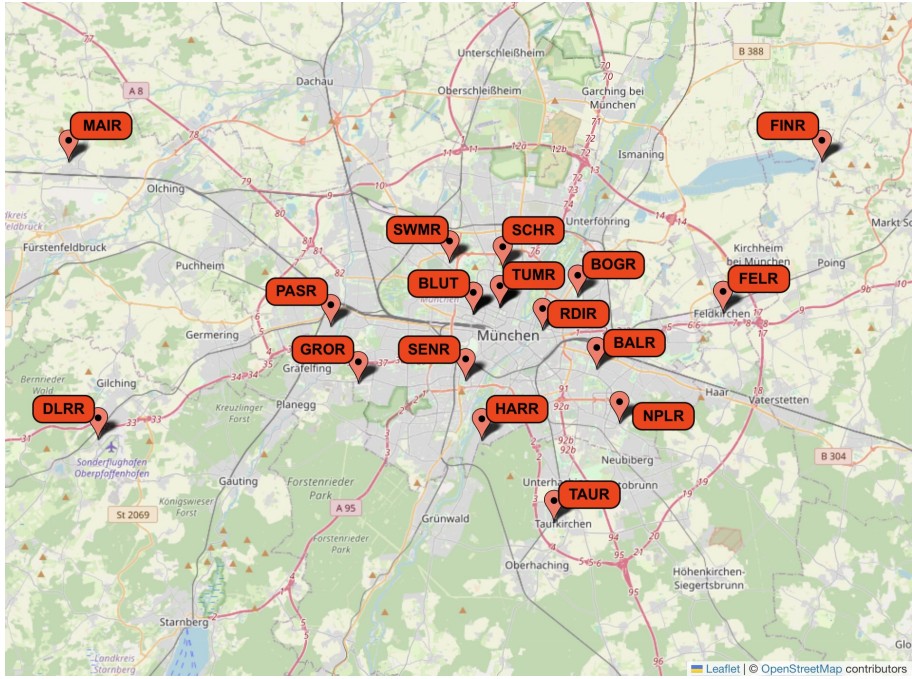

**Figure 4.** Deployment map of the ACROPOLIS network in the Munich metropolitan area. The map shows all seventeen deployment sites, covering urban, suburban, and rural environments. Each site hosts one measurement system, with the exception of TUMR and BLUT, which each operate two systems to support reference comparisons and vertical profiling, respectively.

wind directions. A particularly notable site is the Blutenburg Tower site, which has two systems with inlets at 48 m and 85 m above ground level, allowing for vertical profile measurements.

On the TUM campus, the generation 1 and the generation 2 ACROPOLIS system are deployed along with a reference Picarro
G2401 instrument to evaluate the long-term performance and temporal stability of the updated temperature stabilization.

Detailed information on deployment sites can be found in Table 1 and on site profiles on the ICOS Cities Portal (ICO, 2025).

### 2.5.2 Operation

All stations are connected via LTE and support remote management, including software updates, configuration changes, and system diagnostics. Our modular network architecture allows for easy future expansion and quick changes of selected sites.
Routine site visits are conducted annually to replace calibration gas tanks, verify physical integrity, and inspect inlet and wind sensor positioning. Operational costs are kept low, with approximately 10 € per year per station for mobile communication, 600 € year per station for calibration gas (assuming two tanks per station), and a total of 400 € per year for the shared cloud infrastructure supporting backend services.



**Table 1.** Site classifications for the 17 ACROPOLIS network locations. Entries are sorted by deployment date. Each site hosts one measurement system, except for TUMR and BLUT, which each have two systems installed.

| System Name | Inlet AGL (m) | Inlet ASL (m) | Building Usage | Site Type | Deployment |
|---|---|---|---|---|---|
| TUMR | 31.0 | 542.4 | Research | Urban | 12-01-2024 |
| MAIR | 15.0 | 527.0 | School | Rural | 08-02-2024 |
| PASR | 17.9 | 546.0 | School | Suburban | 08-02-2024 |
| TAUR | 17.8 | 580.4 | School | Rural | 14-02-2024 |
| GROR | 30.4 | 572.4 | Research | Suburban | 14-02-2024 |
| FELR | 15.7 | 539.7 | City Hall | Rural | 22-02-2024 |
| FINR | 15.2 | 509.1 | City Hall | Rural | 22-02-2024 |
| DLRR | 22.9 | 599.0 | Research | Rural | 28-02-2024 |
| SENR | 15.3 | 541.6 | School | Urban | 29-02-2024 |
| RDIR | 20.1 | 543.3 | Hospital | Urban | 15-03-2024 |
| SCHR | 17.1 | 527.6 | School | Urban | 11-04-2024 |
| SWMR | 27.6 | 536.0 | Company | Urban | 14-06-2024 |
| BLUT | 101.6 | 620.2 | Broadcast | Urban | 23-06-2024 |
| NPLR | 31.4 | 571.3 | Hospital | Suburban | 26-06-2024 |
| BOGR | 39.4 | 551.4 | Hospital | Suburban | 09-07-2024 |
| HARR | 29.3 | 582.4 | Hospital | Suburban | 30-07-2024 |
| BALR | 18.1 | 546.1 | School | Suburban | 09-10-2024 |

## 3 Results

This section presents results of the ACROPOLIS network. The primary goals of the network were to achieve a sensor accuracy within a target RMSE of 1 ppm and to capture spatially resolved intra-city $CO_2$ variability. We first assess individual sensor performance through side-by-side comparisons with a Picarro reference instrument, and evaluate calibration and stabilization strategies designed to achieve this accuracy goal with minimal pre-deployment workload. Subsequently, we examine the network's capability to maintain reliable long-term operation and resolve neighborhood-scale variations in $CO_2$.

### 3.1 Side-by-side Setup

The measurement performance of the 20 systems was assessed through side-by-side comparisons with a Picarro G2301 reference instrument (ICOS ID 413) from January to April 2024 at the TUM campus in Munich. A second assessment was conducted for two systems from February to June 2025 using a Picarro G2401 (ICOS ID 529), comparing the performance of a first-generation system (v1, ID 6) with a second-generation system (v2, ID 3) at the same location.





All side-by-side comparisons were carried out under field conditions, with the systems placed on the TUM university rooftop
in their respective outdoor enclosures. The minimum target deployment duration for all systems was two weeks (336 h), with
the longest deployment lasting 85 days (2043 h).

All systems and the reference instrument were connected to individual air inlets placed in close proximity, ensuring that each
system sampled the same ambient air. Due to limitations in power availability, not all systems could be operated simultaneously.

The systems were therefore gradually rotated in and out of operation to allow sequential performance evaluation. Although all
systems experienced a broad range of ambient conditions, some systems were exposed to more extreme conditions than others,
such as higher temperatures and precipitation. More information on ranges can be seen in the scatter plots for sensor sensitivity
in Appendix B. The calibration procedure is the same as for the field deployment and is performed every other day at 03:00
local time using an automated 2-point calibration procedure, as described in Section 2.3.3.

The measurements were processed using a uniform software pipeline described in Section 2.2.3 and 2.3. The Picarro ref-
erence is processed manually. Both data sets are joined and aggregated to hourly means to assess the performance of each
system.

## 3.2 Sensitivity to Environmental Parameters

We rely exclusively on the internal correction functions (Section 2.3.1) of the GMP343 sensor, while minimizing the range of

present environmental conditions through system stabilization. According to the manufacturer, humidity, pressure and oxygen
corrections utilize generic models derived from internal R&D projects, whereas temperature sensitivity parameters are indi-
vidually determined during sensor production. The sensor also internally corrects for the pressure-broadening effect of $H_2O$
molecules on the $CO_2$ absorption spectrum at 4.3 μm, as previously verified through laboratory tests in Grange et al. (2025).

In Fig. 5, we selected two systems (ID 6, 9) that show two examples of detected trends for dependency on environmental

conditions. These scatter plots show the hourly mean difference from the reference Picarro instrument and the range of envi-
ronmental conditions measured by the Bosch BME280 (pressure), SHT45 (humidity), and the internal GMP343 temperature
sensor. The black dotted line indicates perfect agreement between the sensor and reference system and the red line is a linear fit
through all data points to indicate possible linear trends. The results for all 20 sensors can be found in Appendix B. Post-quality
control revealed irregularities in the performance of the SHT45 humidity sensor in system 17. However, these measurements

were not excluded to ensure full transparency. Since the sensitivity plots were not obtained in isolated climate chamber condi-
tions, some degree of cross-sensitivity between parameters is to be expected. The results for the pressure-sensitivity correction
confirm the findings of Shusterman et al. (2016) who observed the pressure dependence to be quite robust between sensor
units. Corrections for humidity show small trends for some sensors, but are overall stable across all systems. We found that
trends in humidity sensitivity correlate with trends observed from temperature sensitivity. Although we find the results from

the internal pressure and humidity correction to be sufficient for our use case, the situation is different for the temperature-
sensitivity correction. The results in Fig. 5 and Appendix B3 show that the temperature sensitivity is highly variable between
sensors, with some sensors showing a weak temperature dependence and others a strong temperature dependence. This is in line





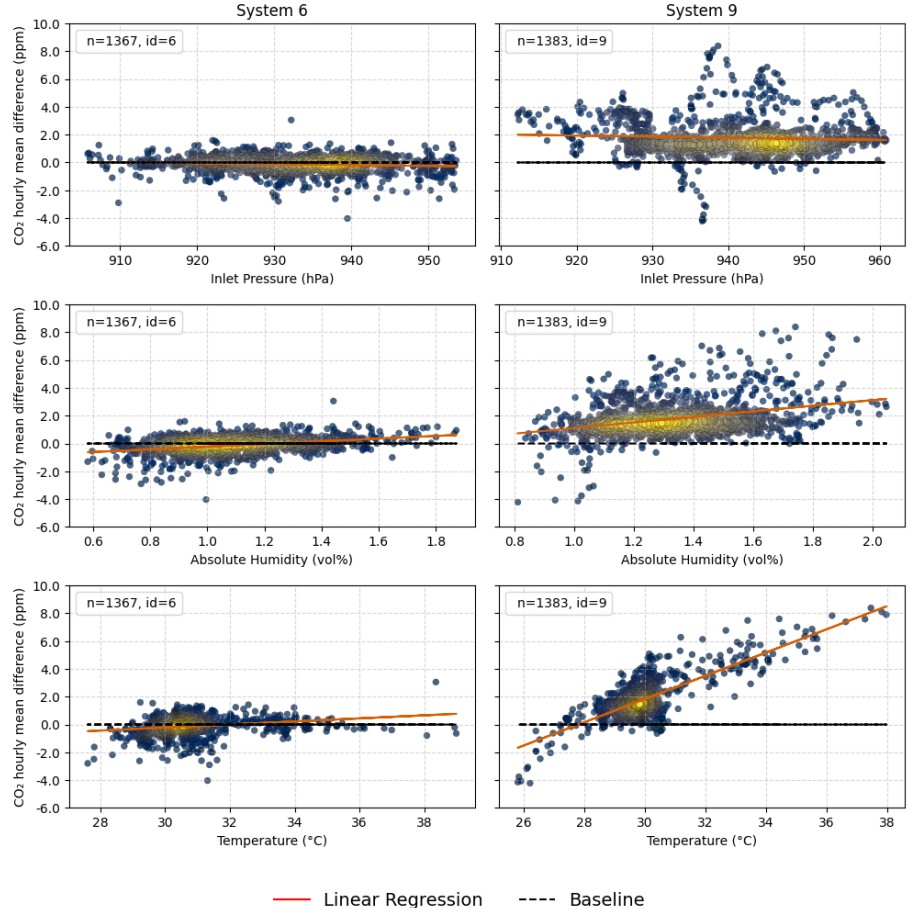

**Figure 5.** Results for sensor sensitivity derived from the 2024 side-by-side comparisons with a Picarro G2301 reference instrument. The scatter plots show the hourly mean difference between two first-generation systems (ID 6 and 9) and the reference instrument (y-axis), plotted against environmental variables: pressure, absolute humidity, and sensor temperature (x-axis). Point color indicates data density, with yellow representing high density and blue low density. The total number of hourly observations is indicated in the top-left corner of each subplot.

with the findings of Delaria et al. (2021), who also observed a strong and in some cases nonlinear sensor-specific temperature dependence.

## 3.3 Sensor Performance Assessment

### 3.3.1 Performance of System Generation 1

Appendix A shows that $R^2$ for all systems is $\geq 0.99$, which indicates a good agreement for the sensor measurements and the reference instrument within the measured range. The results of the sensor performance (MAE, RMSE) are shown in Fig. 6.



The figure shows that the sensors achieve an accuracy of at least 1.1 ppm RMSE, with the exception of systems with a strong temperature sensitivity (ID 4,8,9,10,11,15,17,19) indicated by the red dots. These sensors show a range of 1.1 to 2.6 ppm RMSE. System 17 is the system with the worst performance with an RMSE of 2.6 ppm, which is partially related to the damaged SHT45 sensor. The difference in performance based on temperature sensitivity is magnified in summer conditions, when solar radiation heats up the outdoor enclosure.

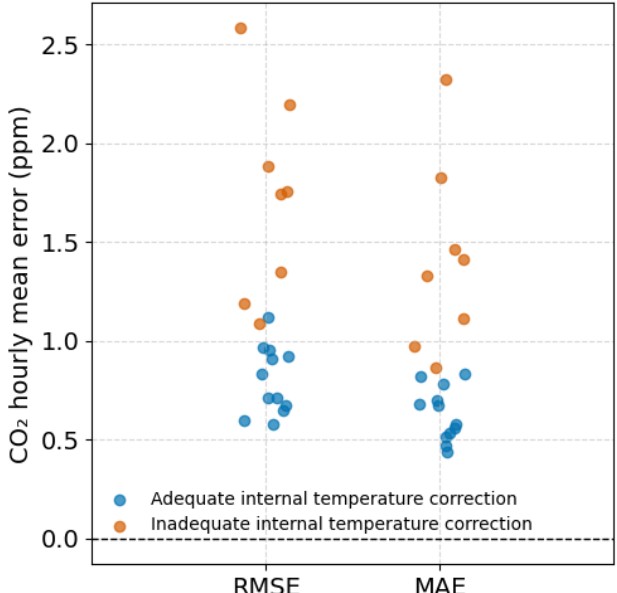

**Figure 6.** Performance metrics from the 2024 side-by-side comparison of first-generation ACROPOLIS systems. The plot shows root mean square error (RMSE) and mean absolute error (MAE) for each sensor compared to a Picarro G2301 reference instrument. Blue and red colors indicate the effectiveness of internal temperature compensation, with red marking sensors that show inadequate correction and heightened sensitivity to temperature variations.

Based on these results, we developed a second-generation prototype featuring an additional temperature-controlled enclosure around the GMP343 sensor. This allows for more precise regulation and a higher temperature target, helping to eliminate heat build-up effects during summer and reducing the sensor's exposure to ambient temperature variability. We kept the 350 W cabinet heater for the outdoor enclosure and added a dedicated PID-controlled temperature-stabilized sensor box (Section 2.1.4) that maintains the sensor environment at the target temperature ±0.1 °C, while only requiring 10 W additional energy input. After assembling the prototype, we performed a second side-by-side comparison for the second-generation system.

### 3.3.2 Performance of System Generation 2

Sensor 3 was one of the first systems deployed during the project, installed at the FINR site in late February 2024. Following a modem hardware failure, it was returned to the laboratory. Being among the remaining sensors at TUM in November 2024,





**Table 2.** Summary statistics from the 2024 side-by-side comparison of the ACROPOLIS network with a Picarro G2301 reference instrument. The table reports the number of hourly measurements, mean bias, mean absolute error (MAE), root mean square error (RMSE), coefficient of determination ($R^2$), and Min-Max ranges for inlet pressure (p), absolute humidity (AH) and sensor temperature (T) for all 20 first-generation systems.

| System ID | n | Mean bias | MAE | RMSE | $R^2$ | Min-Max $p$ (hPa) | Min-Max $AH$ (vol%) | Min-Max $T$ (°C) |
|---|---|---|---|---|---|---|---|---|
| 1 | 478 | 0.28 | 0.83 | 1.12 | 0.995 | $913 - 959$ | $0.56 - 1.56$ | $27.2 - 30.0$ |
| 2 | 839 | 0.41 | 0.57 | 0.71 | 0.998 | $913 - 960$ | $0.64 - 1.83$ | $25.9 - 37.5$ |
| 3 | 854 | 0.01 | 0.67 | 0.90 | 0.998 | $911 - 959$ | $0.39 - 1.64$ | $26.0 - 31.0$ |
| 4 | 1392 | 1.24 | 1.46 | 1.88 | 0.991 | $918 - 960$ | $0.59 - 1.86$ | $26.5 - 37.0$ |
| 5 | 454 | 0.10 | 0.68 | 0.83 | 0.997 | $913 - 960$ | $0.61 - 1.62$ | $28.7 - 30.7$ |
| 6 | 1367 | -0.15 | 0.44 | 0.57 | 0.998 | $905 - 953$ | $0.58 - 1.87$ | $27.6 - 38.9$ |
| 7 | 1343 | 0.77 | 0.82 | 0.96 | 0.998 | $913 - 961$ | $0.64 - 1.93$ | $27.3 - 38.4$ |
| 8 | 670 | 1.03 | 1.41 | 1.75 | 0.990 | $918 - 960$ | $0.61 - 1.63$ | $25.3 - 30.1$ |
| 9 | 1383 | 1.76 | 1.82 | 2.19 | 0.992 | $912 - 960$ | $0.81 - 2.04$ | $25.8 - 37.9$ |
| 10 | 2017 | -0.51 | 0.86 | 1.08 | 0.996 | $912 - 961$ | $0.31 - 1.39$ | $23.8 - 31.7$ |
| 11 | 2041 | 0.92 | 1.33 | 1.74 | 0.992 | $909 - 959$ | $0.35 - 1.71$ | $24.1 - 35.4$ |
| 12 | 715 | -0.37 | 0.56 | 0.71 | 0.999 | $913 - 957$ | $0.36 - 1.53$ | $24.8 - 29.5$ |
| 13 | 385 | 0.23 | 0.69 | 0.95 | 0.998 | $913 - 958$ | $0.37 - 1.59$ | $28.3 - 30.4$ |
| 14 | 1079 | 0.60 | 0.78 | 0.92 | 0.997 | $912 - 960$ | $0.57 - 1.93$ | $27.6 - 37.2$ |
| 15 | 1223 | 0.79 | 0.97 | 1.19 | 0.996 | $913 - 962$ | $0.54 - 1.87$ | $23.2 - 35.8$ |
| 16 | 561 | 0.04 | 0.46 | 0.60 | 0.999 | $917 - 962$ | $0.38 - 1.58$ | $25.8 - 29.9$ |
| 17 | 455 | 2.32 | 2.32 | 2.58 | 0.994 | $917 - 957$ | $0.81 - 2.06$ | $27.6 - 38.0$ |
| 18 | 571 | -0.01 | 0.51 | 0.65 | 0.999 | $916 - 961$ | $0.48 - 1.67$ | $26.5 - 29.6$ |
| 19 | 575 | 0.98 | 1.11 | 1.34 | 0.996 | $912 - 960$ | $0.78 - 2.06$ | $26.2 - 37.0$ |
| 20 | 715 | -0.17 | 0.53 | 0.67 | 0.999 | $911 - 960$ | $0.35 - 1.41$ | $24.5 - 30.0$ |
| Mean | 955.85 | 0.51 | 0.94 | 1.16 | 0.996 | - | - | - |

Sensor 3 showed the strongest temperature sensitivity and was therefore selected for reassembly and integration into the generation 2 system.

Due to its early deployment, Sensor 3 did not participate in the April 2024 side-by-side comparison during warmer conditions. As a result, the v1 data set for this sensor is limited to January and February 2024, when the internal sensor temperatures remained below 31 °C.

Fig. 7 shows the sensor temperatures for the side-by-side comparison period 2025 for the first-generation system (v1, ID 6) and the second-generation system (v2, ID 3). Both systems are located on the TUM rooftop next to each other and should experience comparable conditions. Furthermore, the outside temperature, measured by the LMU Meteo Station MUM01 (450



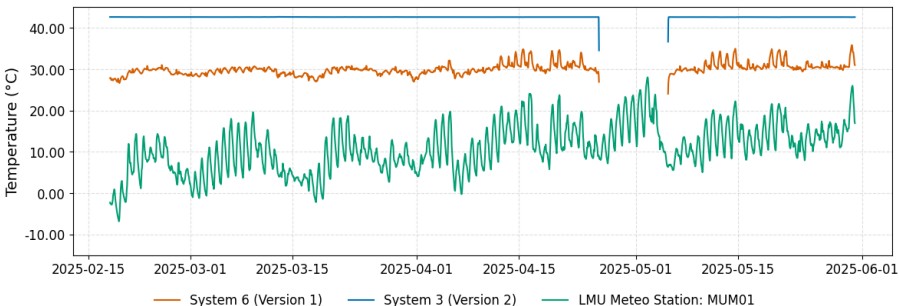

**Figure 7.** Time series of sensor temperature during field deployment at TUMR (February–June 2025). The plot shows internal GMP343 temperature readings from a first-generation (red) and a second-generation (blue) ACROPOLIS system, alongside outside temperature measured at 30 meters by a nearby meteorological station (green). The generation 2 system exhibits stable temperature control and no apparent response to solar radiation, in contrast to the more variable readings from the generation 1 system. The gap in the time series shows a power outage.

meter direct distance) at 30 meters above ground, is shown. The figure shows that the temperature control for generation 2 works as expected, while the temperature for generation 1 is affected by solar radiation in the months April and May. The data gap is a result of a power outage.

Fig. 8 shows the performance of Sensor 3 in both the first-generation (v1, 2024) and the second-generation (v2, 2025)
system versions. The sensor response to pressure and absolute humidity in generation 2 shows trends similar to those in generation 1, but with a visibly reduced spread and lower standard deviation. The stabilized temperature range of 42.5 to 42.7 °C in generation 2 minimizes the influence of temperature on sensor performance, and the temperature sensitivity observed in generation 1 is no longer evident. For this comparison period, the target temperature was set at 40 °C. The observed temperature of 42.6 °C is attributed to activated internal optical heating.

With these hardware upgrades, we could successfully achieve an improvement in sensor accuracy even in the presence of a broader range of environmental conditions for the second-generation system. In total, sensor 3 in the second generation performed with an RMSE of 0.60 ppm, an MAE of 0.49 ppm, and a standard deviation of 0.52 ppm, which is a significant improvement compared to the performance of sensor 3 in the first generation with an RMSE of 0.91 ppm, an MAE of 0.68 ppm, and a standard deviation of 0.91 ppm. The generation 2 data set contains more than 2.5 times the number of data
points and was exposed to a wider range of environmental conditions compared to the v1 data set, highlighting the observed performance improvements.

Although the target temperature of 40 °C worked well for the first half of 2025 the target temperature should be chosen to match local conditions and temperatures. We are currently experimenting with different temperature points for the hot summer months and first results are shown in Appendix E.





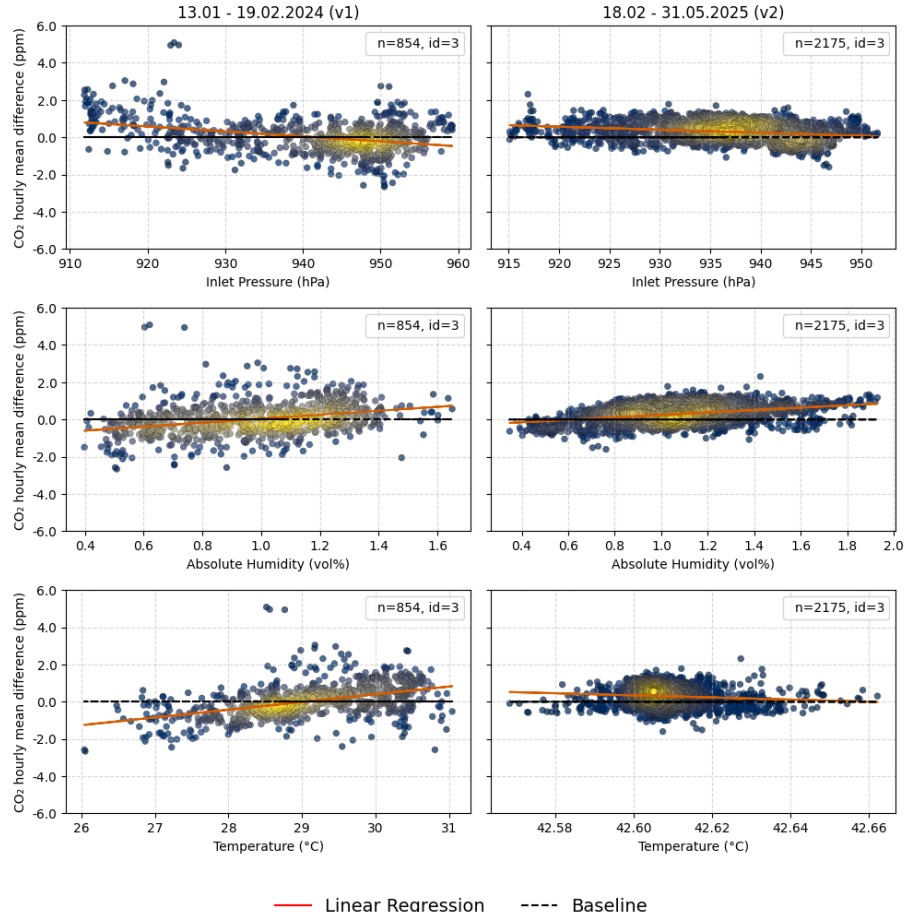

**Figure 8.** Sensor sensitivity results of the side-by-side comparison for the sensor (ID3) in the first-generation system (v1, 2024) and the second-generation system (v2, 2025). Scatter plots show the hourly mean difference between ACROPOLIS systems and the reference instrument (y-axis), plotted against pressure, absolute humidity, and sensor temperature (x-axis). The temperature plots for both generations display different ranges, each representative of their respective stable control regime. Point color indicates data density, with yellow representing high density and blue low density. The total number of hourly observations is displayed in the top-right corner of each subplot. The second-generation system exhibits visibly reduced spread and lower temperature sensitivity compared to the first-generation system.

## 3.4 1-Point Versus 2-Point Calibration Correction

Based on the 2-point calibration data collected during the 2024 side-by-side comparison, we evaluated the performance of a 1-point calibration correction strategy compared to a 2-point calibration correction strategy. The results are shown in Fig. 9. The y-axis shows the difference for the 20 systems between the RMSE calculated from the 2-point calibration correction and the RMSE calculated from the 1-point calibration correction at two different calibration points (400 and 520 ppm). Since the 2-point calibration is anticipated to yield superior performance, the difference between the 1-point and 2-point calibration (1P



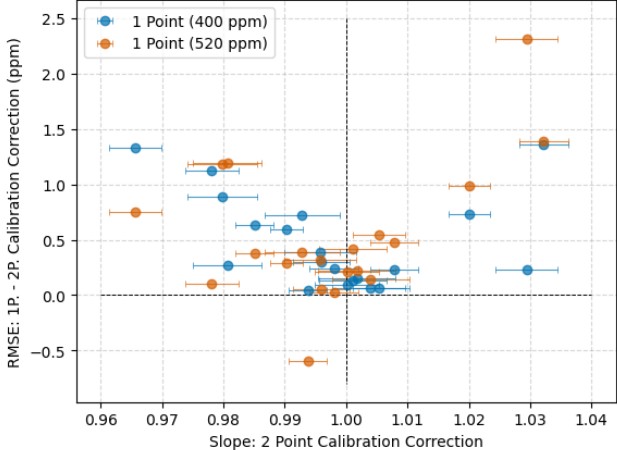

**Figure 9.** Evaluation of 2-point versus 1-point calibration performance. The plot compares the difference in RMSE between the standard 2-point calibration correction and 1-point calibration correction applied at 400 ppm and 520 ppm for all 20 ACROPOLIS systems (y-axis). The x-axis shows the slope derived from the 2-point calibration. Horizontal bars indicate the standard deviation of the interpolated slope across calibration events. The results demonstrate that sensors with slopes close to 1.00 show minimal performance difference, while deviations lead to increased RMSE in the 1-point correction.

- 2P) is expected to be positive. The x-axis shows the slope calculated from the 2-point calibration correction (Section 2.3.3). The horizontal range at each point indicates the standard deviation for the interpolated slope of all performed calibrations. At a slope of 1.00, the 1-point calibration correction becomes mathematically equivalent to the 2-point correction ($y = x + b$). For slopes deviating from 1.00, we expect the performance of the 1-point correction to degrade, resulting in reduced correction accuracy compared to the 2-point method.

In our analysis, we confirmed the expected trend, with the RMSE difference increasing for slopes deviating from 1.00. The results show that the 2-point calibration correction is more accurate than the 1-point calibration correction for all sensors, with the exception of one sensor (ID 17), which shows a negative RMSE difference. This can be considered an outlier, as the system has a damaged SHT45 sensor. Although the trend is clear, the difference in RMSE gives the first indication of the expected performance of a 1-point calibration strategy considering an RMSE target of 1 ppm. The results indicate that the 1-point calibration correction is sufficient for sensors with a slope close to 1.00, but for sensors with a slope deviating, the 2-point calibration correction is preferred to achieve the desired accuracy.

We also found that the slope can change for individual systems over time, which can be attributed to sensor drift or changes in environmental conditions. A visualization of all slopes calculated during the evaluation period can be found in the Appendix C. We think that there is an opportunity for more advanced calibration correction strategies, such as calculating the slope at a lower frequency and relying on a 1-point calibration correction for the daily calibration routine. This would reduce calibration gas consumption and operational cost but require a more complex software pipeline and additional testing to ensure that the



correction is sufficient to maintain the desired accuracy over time. Such a strategy is outside the scope of this work, but could be a future research direction.

### 3.5 Hampel Filter Evaluation at Site MAIR

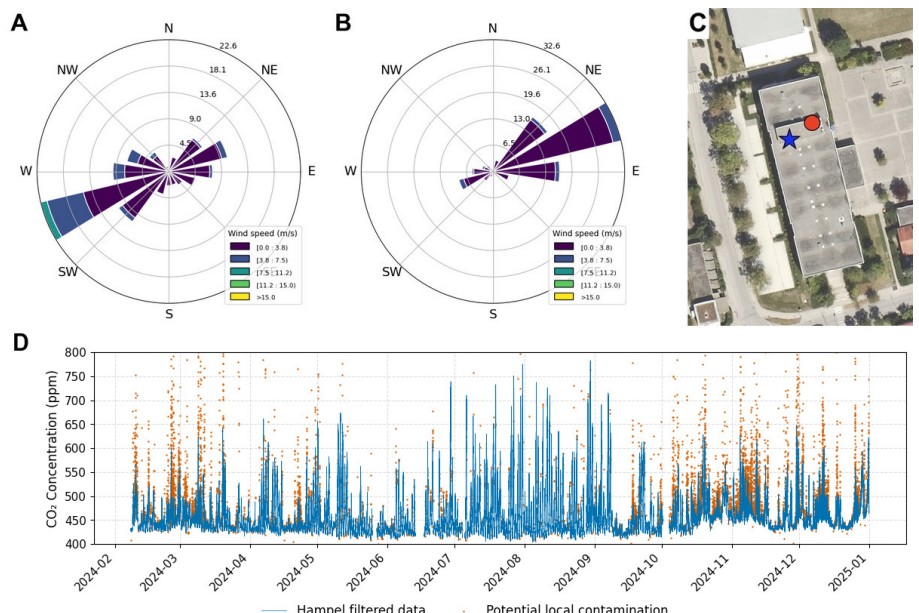

**Figure 10.** (A) Wind speed and direction for measurements without potential local contamination. (B) Wind speed and direction for measurements flagged as potentially contaminated. (C) Aerial view of the school site showing the sensor inlet (blue star) and nearby gas exhaust outlet (red dot) (DOP20RGB imagery by LBDV, CC BY 4.0). (D) Time series of one year of $CO_2$ concentration data (blue) with contaminated observations highlighted in red. Wind roses plotted following (Roubeyrie and Celles, 2018)

With the exhaust of the school gas heating system in close proximity, the MAIR site provides an excellent opportunity to evaluate the performance of the Hampel filter in detecting and removing data affected by local contamination through $CO_2$ sources in close proximity. Located in the northeast of Munich on the northwest edge of the city of Maisach, the inlet is mounted on a former analog antenna pole, while the exhaust outlet is approximately 5 m to the east-northeast (ENE) of the sampling point. Under prevailing southwest wind conditions, Maisach generally acts as a background site.

To assess potential contamination, we applied a Hampel filter (Pearson et al. (2016), El Yazidi et al. (2018)) to 1-minute aggregated $CO_2$ measurements recorded between February 2024 and June 2025. The filter uses a two-hour sliding window and a three-sigma threshold (Section 2.3.4) to identify outliers. For the MAIR site, 2.6 % of the data was flagged and removed as potential local contamination.

Figure 10 (A, B) shows wind rose graphs illustrating the frequency and direction of the winds at the site. Panel A shows all valid data after filtering, highlighting the west-southwest (WSW) as the dominant wind direction. Panel B shows only the



flagged data points, with a clear directional signal from ENE, directly aligned with the location of the exhaust shown in panel C. Most flagged events occur at low wind speeds, suggesting an influence from a nearby source.

The one year time series in panel D shows that flagged contamination events are concentrated in the colder months, particularly in winter, which coincides with the operating period of the heating system.

These results demonstrate that the Hampel filter effectively identifies and removes local $CO_2$ contamination at the MAIR site, particularly during the heating season.

### 3.6 Vertical Profile Measurements at Site BLUT

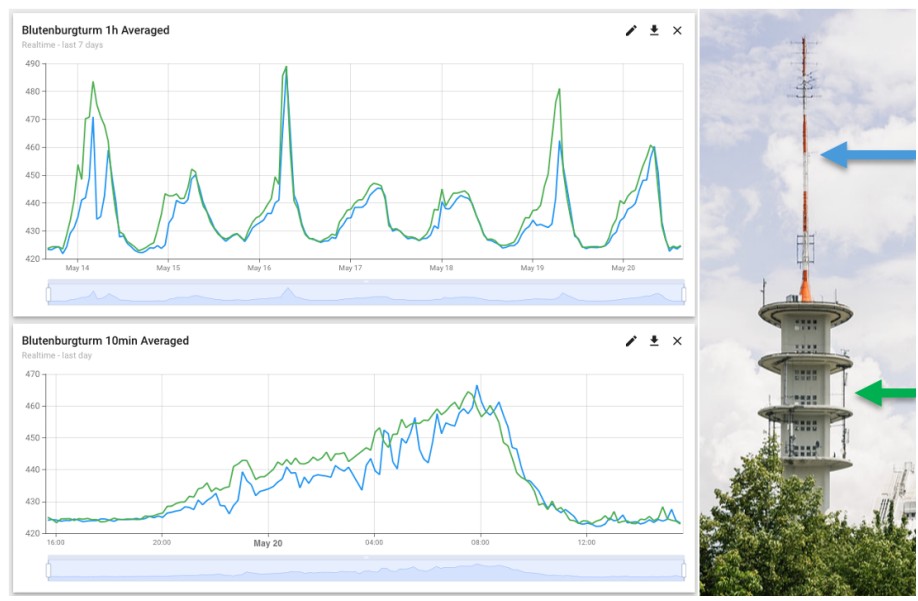

**Figure 11.** Time series of $CO_2$ concentrations at two heights at the BLUT tower for a week in May 2025. Measurements from the ACROPOLIS network are shown for inlets at 48 m (green) and 85 m (blue) above ground level. Data are taken from the live ThingsBoard dashboard and illustrate typical vertical gradients under varying atmospheric conditions. The image shows the measurement inlet positions at the Blutenburg Tower (photo by @ICOS_RI)

Fig. 11 shows two systems deployed in the Blutenburg Tower (BLUT). The inlets are located at 48 m (green arrow) and 85 m
(blue arrow) above ground level (AGL). These systems enable continuous vertical profile measurements of atmospheric $CO_2$ concentrations, allowing detection of vertical gradients. Live data from both inlets are visualized on an interactive dashboard and made publicly accessible via our ThingsBoard instance.

In dense urban environments, nighttime conditions frequently lead to the formation of a stable nocturnal boundary layer, which inhibits vertical mixing. Under such conditions, surface-based $CO_2$ emissions accumulate near the ground, generating
a vertical concentration gradient. (Xueref-Remy et al., 2018) reported that in Paris, vertical $CO_2$ differences were negligible ($\leq 0.1$ ppm) during daytime when the boundary layer is well mixed, but increased to several ppm at night due to suppressed




turbulent mixing. Similarly, (Park et al., 2022) observed nighttime gradients exceeding 20 ppm between 113 m and 420 m in Seoul.

In line with these findings, the ACROPOLIS setup at BLUT regularly resolves $CO_2$ concentration differences between the 48 m and 85 m inlets during nocturnal periods. These gradients are especially pronounced during calm and stable atmospheric conditions, when weak turbulence results in the accumulation of emissions near the surface.

The BLUT site is colocated with additional atmospheric instruments as part of the ICOS-Cities project. Data from the ACROPOLIS system is, for instance, being used as a reference to evaluate a novel method to partition fossil and biogenic contributions to net urban $CO_2$ fluxes using relaxed eddy accumulation (REA) flask sampling (Kunz et al., 2025).

### 3.7 Data Analysis of One Year of Data

Over the course of one year, the Munich ACROPOLIS network collected more than 70 million $CO_2$ measurements across seventeen sites. This dataset enables a high-resolution view of seasonal patterns in urban and rural $CO_2$ concentrations. Seasons are defined meteorologically as spring (March-May), summer (June-August), fall (September-November), and winter (December-February).

To characterize seasonal $CO_2$ patterns across the network, we analyzed diurnal profiles at selected elevated sites, quantified seasonal diurnal variation at all stations, and computed afternoon averages during periods of well-mixed atmospheric conditions.

#### 3.7.1 Diurnal Profiles

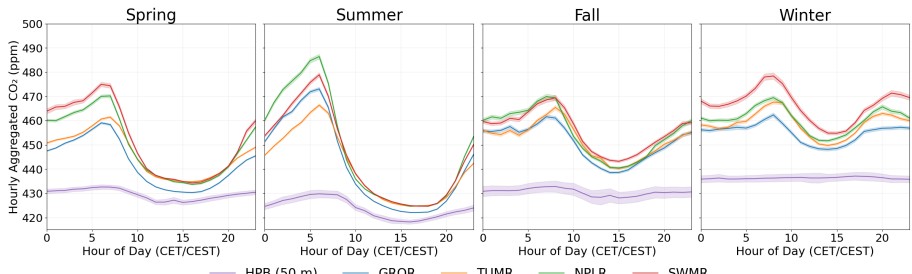

**Figure 12.** Seasonally averaged diurnal $CO_2$ profiles for four ACROPOLIS stations with inlets at 30 m above ground level. The selected stations represent different urban contexts within Munich. For comparison, data from the rural ICOS station Hohenpeißenberg (HPB) are also shown. Shaded areas indicate the 95 % confidence interval.

For a robust assessment of intra-urban $CO_2$ variability and urban–rural gradients, we restricted our analysis to four sites with elevated inlet positions at approximately 30 m above ground level (AGL). These inlets are less affected by immediate surface-level emissions and are more likely to capture air masses representative of the surrounding environment. This approach



helps to ensure that observed differences reflect true spatial patterns rather than differences in measurement height or local source proximity.

The selected sites span different urban contexts across Munich: GROR is located at the southwestern edge of the city and frequently upwind of Munich with respect to the prevailing wind direction, making it a useful reference for incoming background air. TUMR, located in the urban core, serves as a representative high-density inner-city site. SWMR is located in the transition between urban and suburban environments, located between the main inner city highway (Mittlerer Ring) and the busy Dachauer Straße, and is surrounded by mixed land use, including residential areas and green spaces. NPLR, at the southern edge of the city, is adjacent to forested and agricultural land, offering a semi-rural footprint. For additional context and to approximate regional background concentrations, we also include data from the ICOS site Hohenpeißenberg (HPB), located about 50 km southeast of Munich in a rural mountain top location.

Fig. 12 shows the diurnal cycle of $CO_2$ concentrations for each site, averaged per season. The shaded areas indicate the 95 % confidence intervals. Throughout all seasons, GROR exhibits the lowest daytime concentrations among urban sites, consistent with its upwind location and reduced exposure to urban emissions.

In spring and summer, TUMR, SWMR, and NPLR show similar afternoon minima, with overall concentrations lower in summer. This reflects enhanced photosynthetic uptake and higher boundary-layer heights. Nighttime peaks differ by site: In spring, SWMR shows strong early-morning accumulation, likely due to anthropogenic emissions trapped in a shallow boundary layer. In summer, NPLR records the highest nighttime values, likely due to nearby forest and agricultural respiration, while SWMR also shows elevated levels, consistent with its proximity to Olympia Park and Westfriedhof, which are large vegetated areas. TUMR, in contrast, has the lowest nighttime concentrations, likely due to its location in the urban core with less vegetation.

In fall and winter, the diurnal cycle flattens and overall concentrations rise. Reduced biogenic uptake, increased heating emissions, and shallower mixing layers lead to persistently higher $CO_2$. The contrast between urban sites in Munich and the regional background (HPB) is most pronounced in winter. For the urban sites, SWMR exceeds TUMR and NPLR in afternoon $CO_2$, with a distinct late-day peak likely driven by the higher traffic volume in close proximity.

### 3.7.2 Diurnal Variation

To better understand seasonal patterns at all sites, we calculated seasonal diurnal variation. The diurnal variation is the difference between the maximum and minimum diurnal $CO_2$ concentrations. Fig. 13 displays the variation of diurnal $CO_2$ variation for each site and season. The stations are sorted by their summer variation from low to high.

In summer, the largest diurnal variation is observed in rural sites. MAIR and TAUR, both located on the fringe of the city and adjacent to open land, show the strongest variation. In comparison, FELR and FINR are located atop municipal buildings in rural town centers and exhibit lower amplitudes. Suburban stations generally show greater summer variation than central urban sites, consistent with a stronger local biospheric influence. Exceptions are SWMR and SENR, which both show a high summer variation despite their urban classification. This is likely due to their proximity to large parks and green areas, where





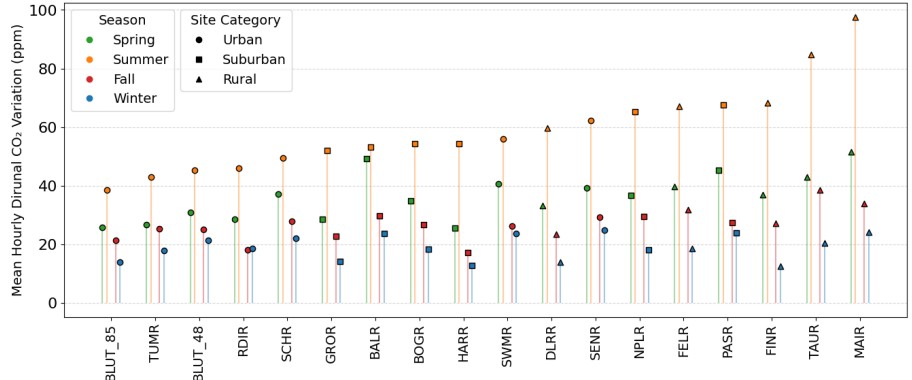

**Figure 13.** Seasonal diurnal variation in hourly aggregated $CO_2$ concentrations across all ACROPOLIS sites. The diurnal variation is defined as the daily maximum minus minimum $CO_2$ concentration. Stations are marked by symbol according to site type: urban, suburban, and rural. The results reveal distinct seasonal and spatial patterns in diurnal $CO_2$ variability.

vegetation can drive both daytime drawdown and nighttime respiration. The lowest summer variation is found at the urban core stations TUMR and BLUT, where limited vegetation leads to a flatter diurnal profile.

In winter, overall diurnal variations are reduced, especially at city edge (e.g., HARR, GROR, NPLR, BOGR) and rural sites, reflecting the absence of daytime photosynthetic uptake and generally shallower boundary layers. An exception is MAIR, which exhibits relatively high winter variation. This is likely due to intermittent influence from a nearby gas heating exhaust 495    located 5 meters east of the sensor inlet, contributing to elevated nighttime or early-morning $CO_2$ peaks. This suggests that Hampel filtering alone may be insufficient to fully remove localized contamination effects from the data.

### 3.7.3    Afternoon Averages

To further assess seasonal differences and spatial variability, we analyzed afternoon (12:00 - 18:00 local time) $CO_2$ concentrations across the network. This period typically reflects the most well-mixed atmospheric conditions and removes the influence 500    of nocturnal accumulation. Fig. 14 shows the seasonal mean for each site in the afternoon, with vertical bars representing the quantiles 2. 5 % and 97. 5 %. The sites are sorted by winter averages from low to high.

The summer and spring averages are consistently lower than in the fall and winter. This seasonal progression is in line with the typical $CO_2$ cycle in urban environments, where biospheric uptake dominates during the growing season, while anthropogenic emissions and limited mixing contribute to winter accumulation.

In winter, rural stations (FINR, FELR, TAUR, MAIR) show lower average concentrations compared to urban and suburban sites. An exception is FELR, which exhibits elevated winter $CO_2$ likely due to its location downwind of the city under prevailing wind conditions. Importantly, rural stations still separate clearly from urban sites despite their lower inlet heights (see Table 1), demonstrating the robustness of the spatial signal. Urban sites show a wider range and particularly high upper quantile values.





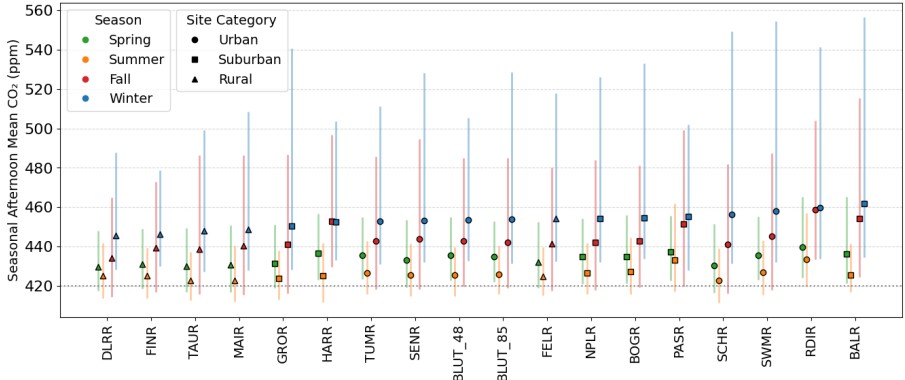

**Figure 14.** Seasonal afternoon $CO_2$ concentrations across all ACROPOLIS sites. Measurements are filtered for local afternoon hours (12:00–18:00), then averaged per season. Vertical bars indicate the 2.5 % and 97.5 % quantiles. The plot reveals clear seasonal patterns in $CO_2$ concentrations and a distinct separation between urban/suburban and rural sites.

In the fall, some sites stand out with unexpectedly high values. PASR and RDIR may be influenced by building ventilation and HARR by ongoing construction activity. We moved the inlet of the RDIR site on late October 2024 to be upwind of the building ventilation outlet, which should reduce local contamination. Optimizing sensor placement is an ongoing process, and further site-specific analysis will enhance network performance.

Taken together, the diurnal variations and afternoon averages reflect the network's ability to resolve spatial differences in $CO_2$ concentrations throughout the city, especially for rural to urban gradients. The seasonal patterns are consistent with expectations based on urban biospheric activity and anthropogenic emissions, while the spatial differences highlight the influence of local sources and sinks. The findings in Munich are consistent with patterns reported in other urban mid-latitude networks (Grange et al. (2025); Xueref-Remy et al. (2018)).

## 4 Conclusions

This study presents the development and the first year of $CO_2$ observations from the Munich ACROPOLIS network, a scalable mid-cost urban greenhouse gas monitoring system.

We explored an alternative sensor correction approach that leverages a combination of manufacturer-provided internal corrections and environmental stabilization within the system. This approach has proven to be effective, as evidenced by side-by-side comparisons between first-generation systems and a high-precision reference instrument. The results demonstrated that the ACROPOLIS mid-cost NDIR sensor network could achieve an average root mean square error (RMSE) of 1.16 ppm with a range of 0.57 to 2.58 ppm, comparable to sensor-specific sensitivity correction strategies derived from pre-deployment characterization (Lian et al., 2024; Grange et al., 2025). Although corrections for pressure and humidity proved to be sufficient across sensors, temperature correction was less straightforward. The results suggest that minimizing the exposure of the sensor to temperature changes is a key to maintaining accuracy. The second generation system implemented a PID-controlled temper-





ature chamber, resulting in significantly improved temperature stabilization. With dedicated temperature control, we believe
that mid-cost Vaisala GMP343 sensors can be utilized to reliably perform around 0.5 to 1.0 ppm RMSE out of the box.

The open-source software solutions developed for ACROPOLIS enable fully automated network operations, on-device data
processing, and publicly accessible live dashboards. Automated device provisioning and remote management significantly
streamline the deployment process for new stations. The post-processing pipeline effectively manages large volumes of data
and offers tools for automated data publication through the ICOS Cities data portal. The system is designed to be scalable, low
maintenance, and reliable, making it a valuable tool for urban greenhouse gas monitoring.

During its first year, the ACROPOLIS network successfully captured distinct seasonal and spatial variations in $CO_2$ concentrations within the metropolitan area of Munich. The network effectively resolved both horizontal and vertical concentration
gradients and measured diurnal cycles influenced by biogenic and anthropogenic activities, and boundary layer dynamics.
These results underscore the capacity of the network to resolve information on the localized characteristics of urban environ-
ments.

Future work will focus on refining existing calibration procedures to reduce the duration of the calibration and explore the
feasibility of less frequent calibration adjustments. Such refinements will further reduce operational costs and maintenance
efforts to support a large number of sensors in the network. In addition, upgrading all first-generation stations to second-
generation hardware standards will ensure consistent and improved performance throughout the network. A key next step
is the integration of collected $CO_2$ data with detailed meteorological information, atmospheric footprint models, and urban
emission inventories. Insights gained here can inform similar deployments in other cities, contributing to a broader effort
toward a harmonized urban greenhouse gas monitoring infrastructure.

*Code and data availability.* L1 and L2 data products (Aigner et al., 2024) are published on the ICOS Cities data portal (https://citydata.icos-
cp.eu/portal/) tagged with the keyword ACROPOLIS. The software developed for this project is available on GitHub: ACROPOLIS-edge
(Aigner et al., 2025), ACROPOLIS-data-processing (Aigner and Chen, 2025a), ThingsBoard-Downloader (Aigner and Chen, 2025b).




## Appendix A: Scatter Plots

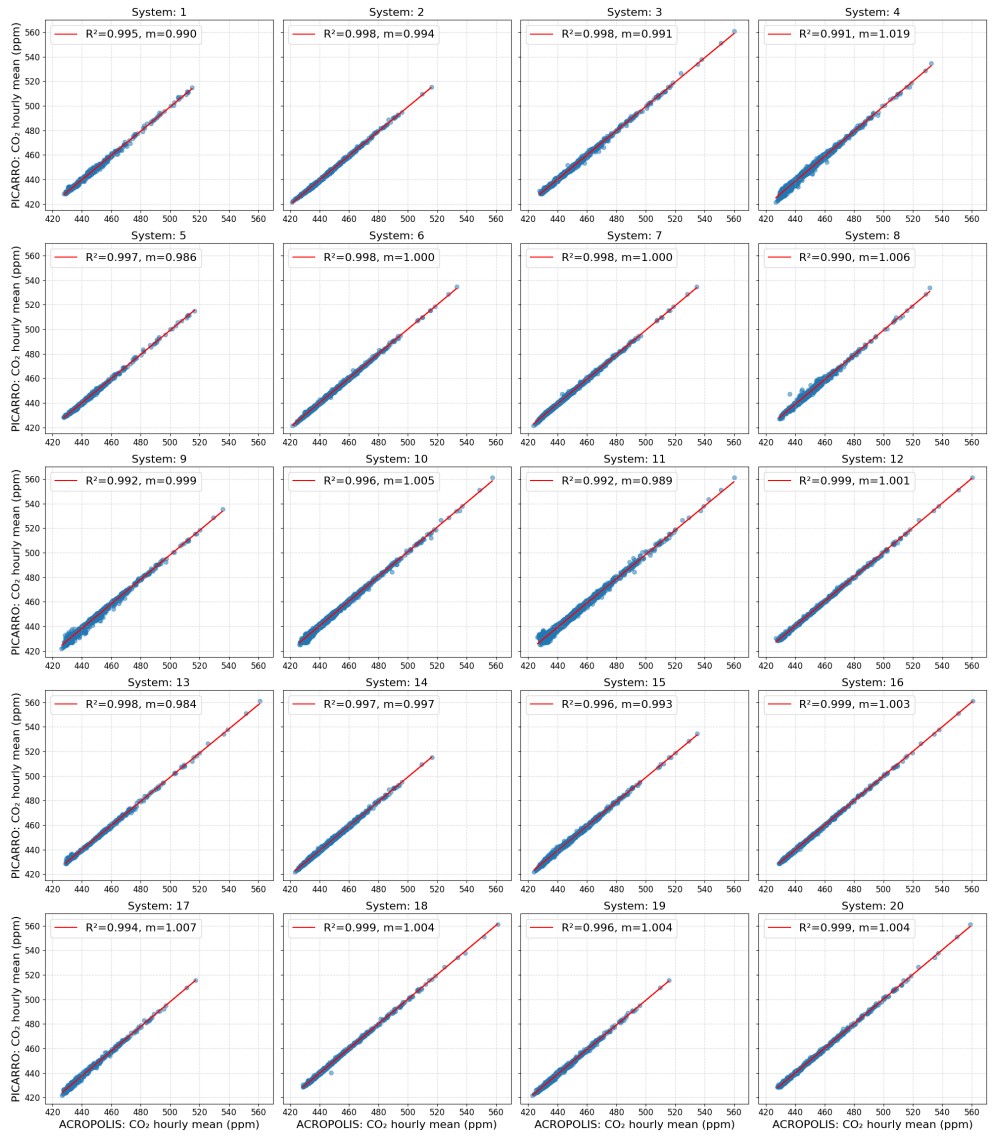

**Figure A1.** Scatter plots of hourly averaged $CO_2$ concentrations measured by all twenty ACROPOLIS generation 1 systems compared to a Picarro G2301 reference instrument during the 2024 side-by-side evaluation. The coefficient of determination ($R^2$) and slope ($m$) are shown in the top left of each subplot. Differences in the concentration range across subplots reflect the respective deployment periods of the individual systems.





## Appendix B: Sensitivity Scatter Plots

### B1  Pressure

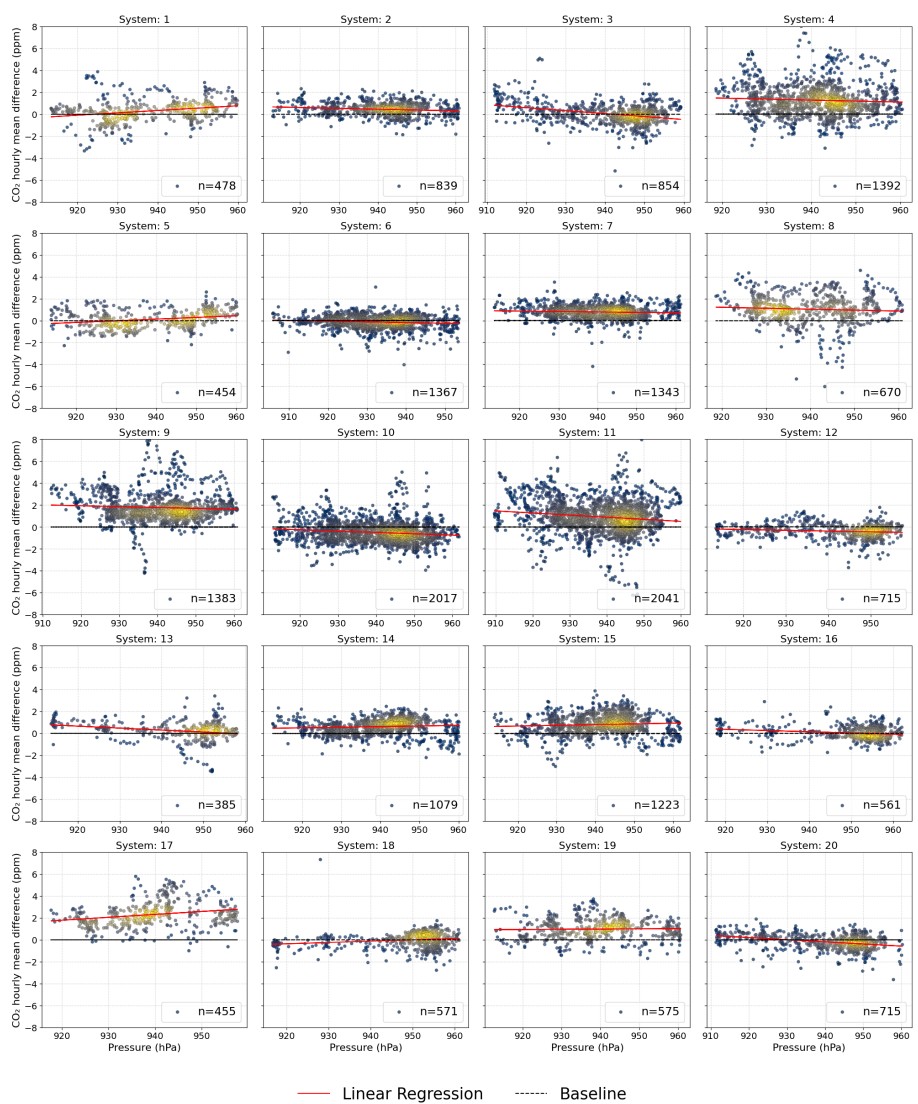

**Figure B1.** Pressure sensitivity analysis of hourly averaged $CO_2$ concentration differences between each of the twenty ACROPOLIS generation 1 systems and a Picarro G2301 reference instrument during the 2024 side-by-side evaluation. The x-axis shows ambient pressure measured by the BME280 sensor. Pressure ranges vary depending on the system's deployment period. Point color represents data density, with yellow indicating high density and blue low density. The total number of hourly observations is noted in each subplot legend. No systematic trends are observed across the range of ambient pressures.



## B2  Absolute Humidity

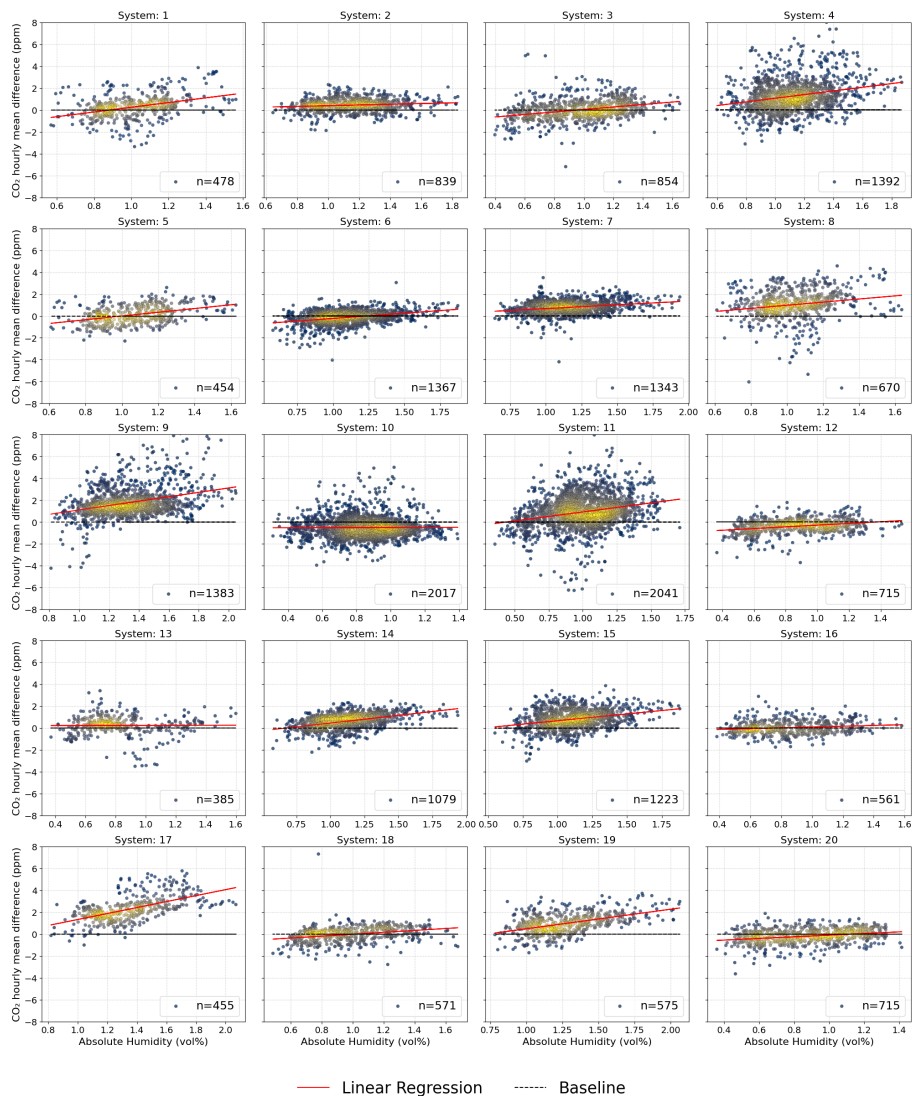

**Figure B2.** Sensitivity of hourly averaged $CO_2$ concentration differences to absolute humidity, measured between each of the twenty ACROPOLIS generation 1 systems and a Picarro G2301 reference instrument during the 2024 side-by-side evaluation. The x-axis shows ambient absolute humidity, calculated from SHT45 relative humidity, BME280 pressure, and GMP343 temperature measurements. Absolute Humidity ranges vary depending on the system's deployment period. Point color indicates data density, with yellow representing high density and blue low density. The total number of hourly observations is shown in each subplot legend. Several systems display systematic trends across the absolute humidity range, which appear to correlate with temperature sensitivity.





## B3 Temperature

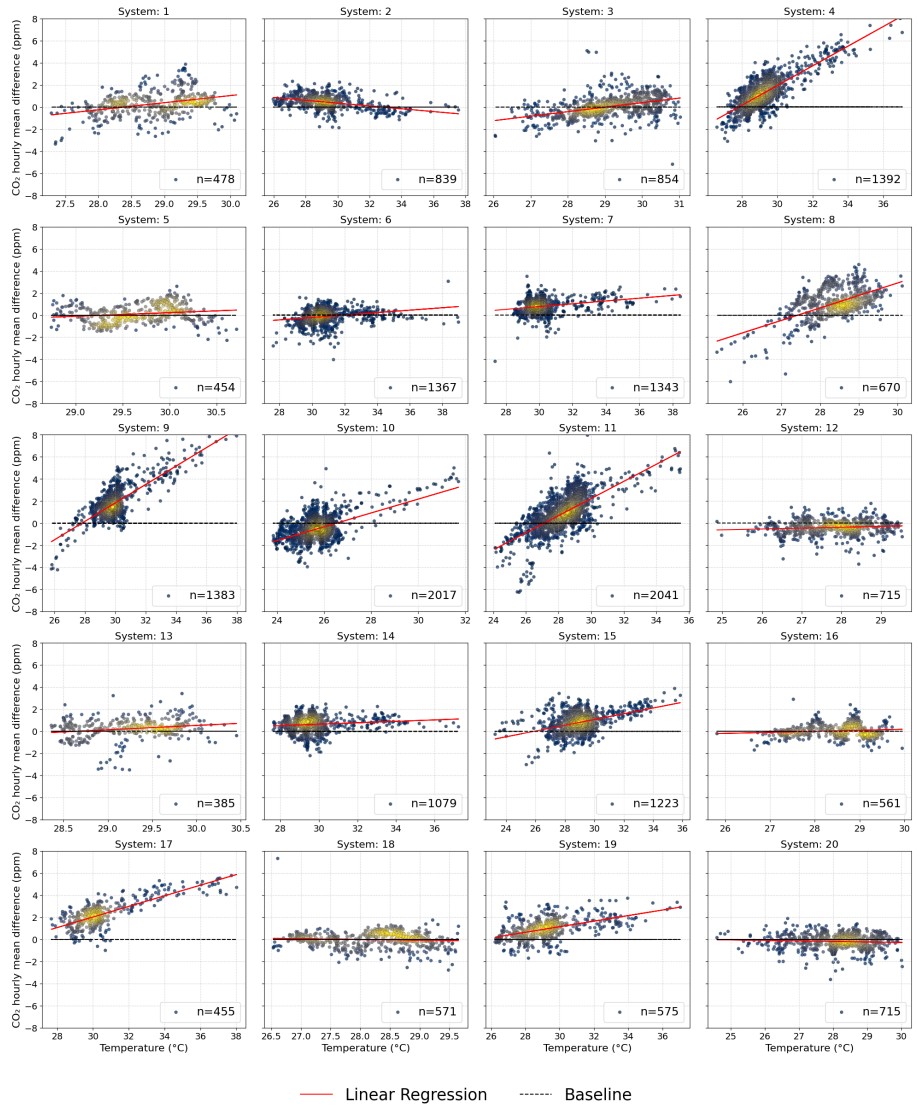

**Figure B3.** Sensitivity of hourly averaged $CO_2$ concentration differences to sensor temperature, measured between each of the twenty ACROPOLIS generation 1 systems and a Picarro G2301 reference instrument during the 2024 side-by-side evaluation. The x-axis shows ambient temperature as measured by the GMP343 sensor. Sensor temperature ranges vary depending on the system's deployment period. Point color indicates data density, with yellow representing high density and blue low density. The total number of hourly observations is shown in each subplot legend. Systematic temperature-dependent trends are evident in several systems, most notably IDs 4, 8, 9, 10, 11, 15, 17, and 19.



**Appendix C: Slope over Time**

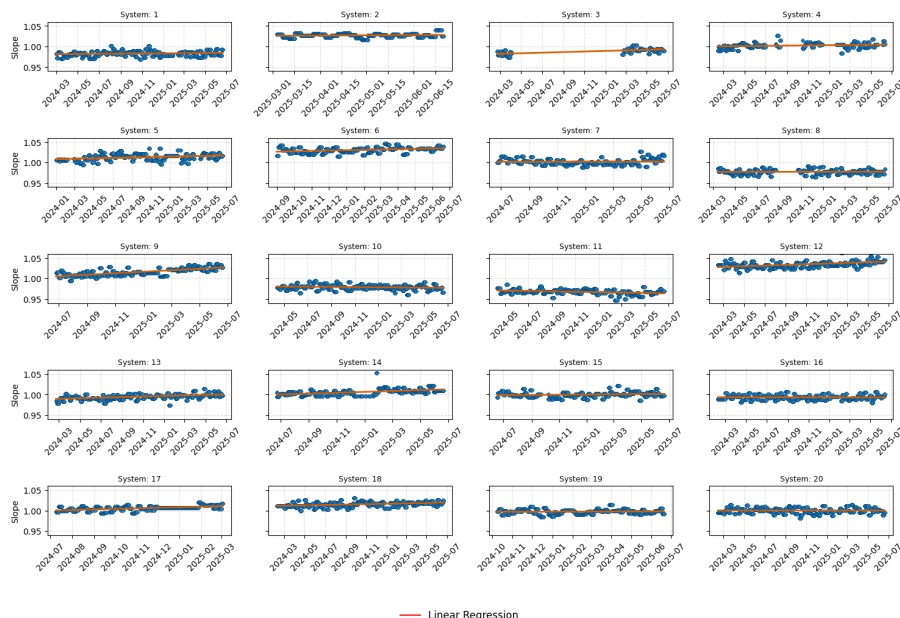

**Figure C1.** Scatter plots of calibration slopes obtained from the 2-point calibration procedure for all twenty ACROPOLIS generation 1 systems across their respective field deployment periods until July 2025. System 3 additionally includes results from the second-generation hardware configuration. The y-axis shows the slope parameter ($m$) derived from each calibration event. The x-axis indicates the corresponding deployment date. No system exhibits pronounced long-term drift, indicating stable calibration behavior across the network.



## Appendix D: ThingsBoard Public Dashboard

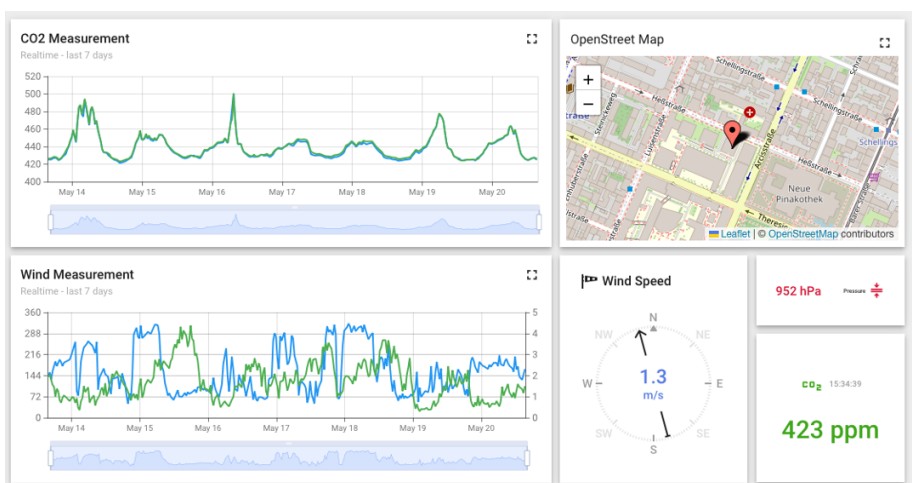

**Figure D1.** Screenshot of the publicly available ACROPOLIS ThingsBoard dashboard. The interface provides an overview for each site, including weekly time series of $CO_2$ concentration, wind speed, and wind direction. Live measurements of wind speed and direction, enclosure pressure, and $CO_2$ concentration are shown alongside. An interactive map in the top right corner displays the geographic location of the selected station. The example shown corresponds to the TUMR site, which includes measurements from both the first- and second-generation systems. A link to the dashboard is available on the ACROPOLIS-edge GitHub page (Aigner et al., 2025).

## Appendix E: Summer Month Performance

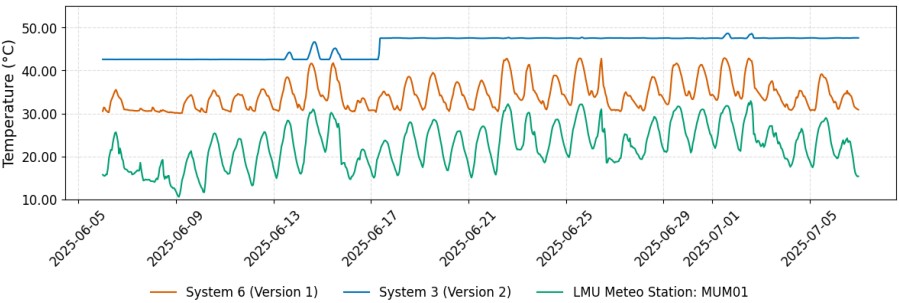

**Figure E1.** Time series of sensor temperature during field deployment at the TUMR site (June–July 2025). The plot shows internal GMP343 temperature readings from a first-generation (red) and a second-generation (blue) ACROPOLIS system, along with ambient temperature measured at 30 meters above ground by a nearby meteorological station (green). The temperature setpoint for the second-generation system was increased to 45 °C to assess its suitability under elevated summer conditions. The higher target temperature results in a more stable and decoupled sensor environment.



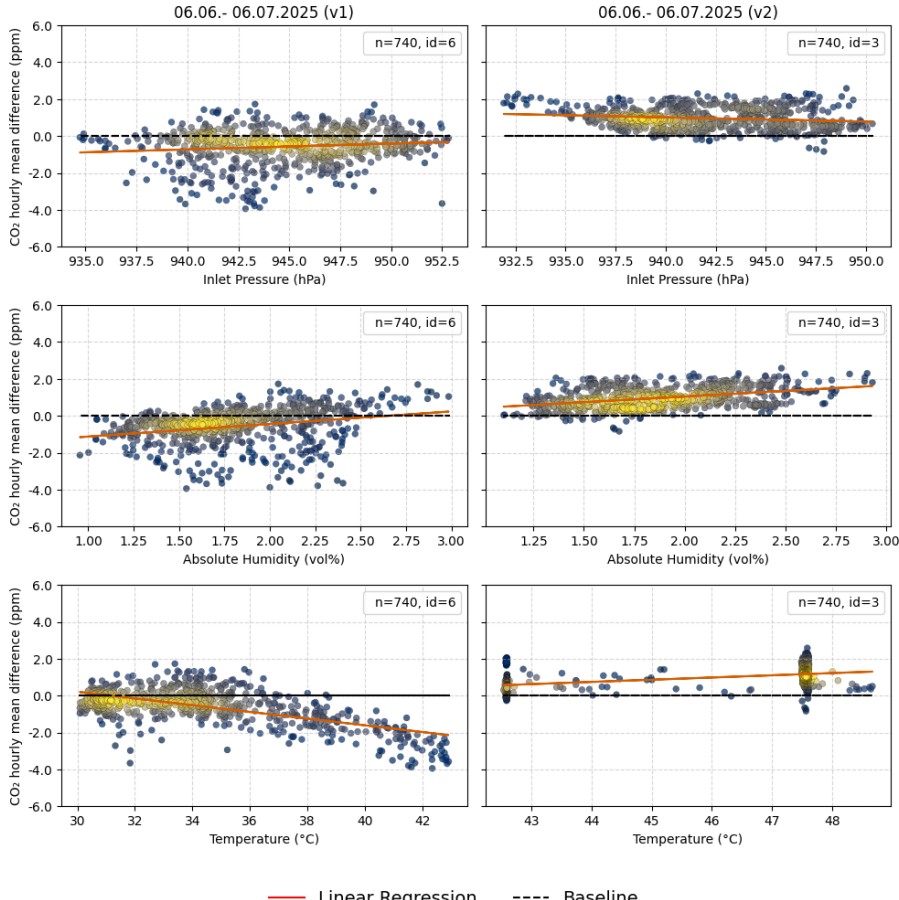

**Figure E2.** Sensor sensitivity results from the side-by-side comparison of two ACROPOLIS systems (IDs 3,6), illustrating the performance of the first- and second-generation configurations during elevated summer temperatures in June–July 2025. Scatter plots show the hourly mean difference between ACROPOLIS and the reference instrument (y-axis), plotted against ambient pressure, absolute humidity, and internal sensor temperature (x-axis). Point color indicates data density, with yellow representing high density and blue low density. The total number of hourly observations is indicated in the top-right corner of each subplot. The first-generation system shows a visible change in temperature sensitivity above 36 °C. In contrast, the second-generation system maintains stable performance at 40 °C and 45 °C, including during brief exceedances of the setpoint. A small bias observed at 45 °C may be related to the temperature dependence of the co-located low-cost sensors (BME280 and SHT45), warranting further investigation.

*Author contributions.* PA, DK, KK, AS built the systems, PA, FB, LF, MM, AS contributed to the software, PA, JC, DK, KK, AS, AW
contributed to the hardware design, PA, DK, KK deployed and operated systems, PA processed and analyzed the data, JC, MC, LE, SG, DK, OL, PR, AW provided scientific advice, PA wrote the manuscript, All authors reviewed the manuscript, JC conceived the concept and supervised the project as the project PI.



*Competing interests.* The authors declare that they have no conflict of interest.

*Acknowledgements.* We gratefully acknowledge the cooperation and support of the managers and technical staff who made station de-
ployment possible: Landeshauptstadt München – Referat für Bildung und Sport, Stadtwerke München GmbH, München Klinik gGmbH,
Gemeinde Feldkirchen, Gemeinde Taufkirchen, Gemeinde Finsing, Gemeinde Maisach, Deutsches Zentrum für Luft- und Raumfahrt (DLR),
Ludwig-Maximilians-Universität München – Referat IV.7, and the Technical University of Munich – Klinikum rechts der Isar. We thank the
ICOS Flask and Calibration Laboratory at the Max Planck Institute for Biogeochemistry (Jena) for providing WMO-traceable calibration
gases. We thank the Deutscher Wetterdienst Meteorological Observatory Hohenpeissenberg for providing a Picarro reference instrument for
the duration of the project. Finally, we gratefully acknowledge Moritz Angleitner and Daniel Stahl for their support in building the systems.
We used ChatGPT-4.5 (OpenAI) and Writefull to improve readability and to check spelling and grammar of the manuscript.

This work has been funded by the ICOS PAUL project: PAUL, Pilot Applications in Urban Landscapes - Towards integrated city observa-
tories for greenhouse gases (ICOS Cities), funded by the European Union's Horizon 2020 Research and Innovation Programme under grant
agreement No 101037319. Furthermore, the work is partly supported by the ERC consolidator grant CoSense4Climate (Grant 101089203).





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
