# Peer review of "ACROPOLIS: Munich Urban CO2 Sensor Network"

_EGUsphere, 2025_

## Author Comment (AC1)

**Response to Anonymous Referee #1 for EGUSPHERE-2025-4157 (ACROPOLIS: Munich Urban CO2 Sensor Network)**

Patrick Aigner1, Jia Chen1, Felix Böhm1, Mali Chariot2, Lukas Emmenegger3, Lars Frölich1, Stuart Grange3,4, Daniel Kühbacher1, Klaus Kürzinger1, Olivier Laurent2, Moritz Makowski1, Pascal Rubli3, Adrian Schmitt1, and Adrian Wenzel1

**Correspondence:** Patrick Aigner (patrick.aigner@tum.de) and Jia Chen (jia.chen@tum.de)

**Anonymous Referee #1**

**General Comment**

Overall this is very good work and consistent with previous findings cited here, that lower cost trace gas sensors have utility if properly calibrated and corrected, but each sensor must be independently corrected as they can vary in their usefulness. I think my only scientific comment (and I do not think it is required to be addressed for publication) is that PBL height was mentioned in passing in Section 3.7.2 e.g. "generally shallower boundary layers". If it's not too much trouble, and if the data exists, it may be good to show some PBL height observations from an urban and rural location nearby to strengthen this argument. It could be purely vegetation based in terms of the diurnal variation, but could it also be due to urban/rural PBL variations too? Just something to consider, overall very nice work!

**10 Response:**

We sincerely thank the reviewer for the positive and encouraging feedback, as well as for the time and effort dedicated to reviewing our manuscript. We agree that differences in planetary boundary layer (PBL) height between urban and rural environments could influence the observed diurnal  $CO_2$  variations and that this aspect would be valuable to investigate in more detail. Unfortunately, we are not aware of any PBL height observations for the general area of Munich that would allow us to make this comparison. For this reason, we consider this analysis to be beyond the scope of the project. We will gladly take this aspect into account in future studies. We removed the statement from the manuscript.

**Removed from the manuscript:** (Page 26, After Line 506)

and generally shallower boundary layers.

<sup>1Technical University of Munich (TUM), Munich, Germany

<sup>2Laboratoire des Science du Climat et de l'Environnement (LSCE/IPSL), Gif-sur-Yvette, France

<sup>3Swiss Federal Laboratories for Materials Science and Technology (Empa), Dübendorf, Switzerland

<sup>4University of Bern, Bern, Switzerland

---

## Author Comment (AC2)

**Response to Anonymous Referee #2 for EGUSPHERE-2025-4157 (ACROPOLIS: Munich Urban CO2 Sensor Network)**

Patrick Aigner1, Jia Chen1, Felix Böhm1, Mali Chariot2, Lukas Emmenegger3, Lars Frölich1, Stuart Grange3,4, Daniel Kühbacher1, Klaus Kürzinger1, Olivier Laurent2, Moritz Makowski1, Pascal Rubli3, Adrian Schmitt1, and Adrian Wenzel1

Correspondence: Patrick Aigner (patrick.aigner@tum.de) and Jia Chen (jia.chen@tum.de)

**Anonymous Referee #2**

**General Comment**

This paper reports the initial results of establishing a  $CO_2$  observation network using the low-cost NDIR-based Vaisala GMP343 sensors at 17 sites across the city of Munich.

The study focuses on two main aspects: (1) how the accuracy of the sensors used in the network was improved, and (2) how the system was applied to capture high-resolution spatial and temporal  $CO_2$  variability within the city.

For the first aspect—sensor accuracy—the study compares the sensitivity of the Vaisala GMP343 sensors to three environmental variables (humidity, pressure, and temperature) against a Picarro reference instrument. Among these variables, temperature had the greatest impact on the NDIR sensors. In the second-generation network, an additional temperature stabilization enclosure was introduced to address this issue. As a result, the RMSE decreased from a maximum of 2.6 ppm in the first-generation system to less than 1 ppm in the second generation, achieving the target accuracy. In summary, the study aimed to enhance NDIR sensor accuracy primarily by controlling the temperature factor.

For the second aspect, the monitoring sites were categorized into three zones—urban, suburban, and rural—to examine spatial variability in CO2 concentrations. At one specific site (MAIR), a Hampel filter was applied to remove the influence of nearby ventilation outlets. Although the filter effectively removed some peaks, it was not entirely successful in eliminating all local pollution signals. The study found that the classified zones showed clear diurnal and seasonal differences: during summer, rural and suburban sites exhibited greater diurnal variability than urban sites due to photosynthetic activity. This pattern persisted in winter, though the diurnal amplitude was considerably smaller.

Overall, the study is well conducted, but several areas require revision or clarification before publication. Please refer to the comments below:

<sup>1Technical University of Munich (TUM), Munich, Germany

<sup>2Laboratoire des Science du Climat et de l'Environnement (LSCE/IPSL), Gif-sur-Yvette, France

<sup>3Swiss Federal Laboratories for Materials Science and Technology (Empa), Dübendorf, Switzerland

<sup>4University of Bern, Bern, Switzerland

**Response:**

We sincerely thank the reviewer for the positive and encouraging feedback, as well as for the time and effort dedicated to reviewing our manuscript. We address the individual comments point by point below.

(1) (Page 6, Line 123) The paper states that the intake line was extended up to 50 m, with a flow rate of about 0.5 LPM. Is this flow rate sufficient for such a long sampling line? Please provide a proper justification or reference.

**Response:**

40

We thank the reviewer for this valuable comment and fully agree that the choice of tubing diameter and flow rate is critical to ensure representative air sampling, particularly when longer intake lines are used. To verify that a 50 m sampling line is suitable for our system, all stations were operated with 50 m of tubing during the 2024 site-by-site evaluation campaign to simulate the most demanding deployment conditions. The site-by-site results confirmed that this configuration is appropriate, as no significant deviations or artifacts were observed compared to the reference measurements.

To further substantiate this, we quantified the flow characteristics for the 50 m of 1/4'' (Outside Diameter (OD): 6.35 mm, Internal Diameter (ID): 4.3 mm) SERTOflex tubing at a flow rate of 0.5 L/min. The corresponding Reynolds number is approximately 160, confirming laminar flow. Under these conditions, the residence time is about 1.45 min, which is acceptable given that we are interested in hourly averaged data rather than high-frequency measurements. The calculated pressure drop of 0.09 hPa is negligible compared to the approximately 15 hPa drop caused by the 2  $\mu$ m inlet filter.

In practice, the installed tubing lengths are, depending on local conditions on the rooftops, between 10 and 20 m, resulting in residence times of only 17–34 s. We updated the manuscript to include the additional information presented here.

**Addition to the manuscript:** (Page 6, After Line 125)**

The choice of tubing diameter and flow rate is important to ensure representative air sampling, particularly when longer intake lines are used. In our configuration, ambient air is drawn through 50 m of 1/4'' (Outside Diameter (OD): 6.35 mm, Internal Diameter (ID): 4.3 mm) SERTOflex tubing at a flow rate of 0.5 L/min. To verify the suitability of this setup, we calculated the Reynolds number, residence time, and pressure drop (See Appendix Appendix F). The corresponding Reynolds number of approximately 160 confirms laminar flow. Under these conditions, the residence time is about 1.45 min, which is acceptable given that the network provides hourly averaged data rather than high-frequency observations. The calculated pressure drop of 0.09 hPa is negligible compared to the approximately 15 hPa pressure drop introduced by the 2  $\mu$ m inlet filter.

In practice, the installed tubing lengths vary depending on local rooftop conditions and typically range between 10–20 m, resulting in residence times of only 17–34 s.

(2) (Page 9, Line 215) Calibration was performed only at two points—400 ppm and 520 ppm—for slope/intercept correction. Can linearity across a wide and long-term concentration range (350–600 ppm) be ensured with only two calibration points? Since actual CO2 levels in different urban zones may fall outside this range, would additional multi-point calibration or slope tracking be necessary?

**45 Response:**

60

We appreciate the reviewer's thoughtful comment regarding the number of calibration points. We agree that including additional calibration points could further enhance the robustness of the calibration across a broader concentration range. However, implementing multi-point calibration increases both the operational complexity and the overall cost of the system, which would reduce the scalability and long-term maintainability of the network. One of the main design goals of ACROPOLIS was to achieve a balance between accuracy, simplicity, and deployability across multiple urban sites. As our study focuses on the well-mixed urban background signal rather than strong local point sources, the chosen calibration points at 400 ppm and 520 ppm were selected to effectively cover the expected range of ambient CO2 concentrations in the studied environment. To substantiate this choice, we analyzed the distribution of measured CO2 concentrations across all urban ACROPOLIS sites in Munich for the period from January 2024 to October 2025 (see Figure 1). The resulting histogram confirms that the vast majority of CO2 observations fall within the 400–520 ppm range, with only a small fraction of data points outside this interval. Moreover, as illustrated by the scatter plot in Appendix A of the manuscript, the applied two-point calibration yields a stable and accurate correction even for measurements slightly beyond the selected range. These results demonstrate that the adopted two-point calibration strategy provides sufficient linearity and accuracy for the intended application while maintaining the scalability required for a scaleable, city-wide sensor network.

Figure 1. Distribution of over 12 Million  $CO_2$  concentrations measurement across all urban ACROPOLIS sites in Munich for data from January 2024 until October 2025. The vertical dashed lines indicate the mean and median. Interquantile range (IQR), kernel density estimate (KDE), and 2.5% and 97.5% percentiles are shown.

Addition to the manuscript: (Page 37)
Added Figure 1 as Appendix G.

(3) (Page 9, Line 203) The use of the Wagner equation to calculate water vapor saturation pressure for deriving dry mole fractions seems appropriate. However, since the water vapor data came from an external instrument, that instrument itself likely has some uncertainty. Would this not affect the accuracy of the dry CO2 mole fraction? Please discuss this potential limitation.

**65 Response:**

We thank the the reviewer and agree that uncertainties in humidity measurements can influence the correction from wet to dry  ${\rm CO_2}$  mole fractions. To assess this, we evaluated the expected impact of the SHT45 humidity sensor's specified accuracy ( $\pm 1$  % RH) under representative environmental conditions. Using the Wagner equation, the resulting effect on the dry  ${\rm CO_2}$  mole fraction is approximately 0.04 ppm under cool and dry conditions (5 °C, 10 % RH), and approximately 0.20 ppm under warm and humid conditions (30 °C, 80 % RH). In all cases, this influence is minor compared to our target performance. Moreover, each system undergoes a bias correction with dry reference gas during calibration, which further minimizes any residual offset. Based on the 2024 site-by-site evaluation campaign, we confirmed that the overall uncertainty of the dry  ${\rm CO_2}$  measurements meets our target performance when compared to the Picarro reference, indicating that the humidity correction does not introduce significant additional uncertainty.

We therefore conclude that the contribution of humidity measurement uncertainty to the calculated dry CO2 values is acceptable for our application.

**Addition to the manuscript:** (Page 10, After Line 223)**

Uncertainties in the humidity measurements can influence the correction from wet to dry  $\mathrm{CO}_2$  mole fractions. To quantify this effect, we evaluated the impact of the SHT45 humidity sensor's specified accuracy ( $\pm 1~\%$  RH) under representative environmental conditions. Using the Wagner equation, the resulting uncertainty in the dry  $\mathrm{CO}_2$  mole fraction is approximately 0.04 ppm under cool and dry conditions (5 °C, 10 % RH) and about 0.20 ppm under warm and humid conditions (30 °C, 80 % RH). Across all relevant conditions, this effect remains negligible compared to the overall target precision of the system.

(4) (Page 10, Line 230) Using a long analysis window may risk classifying short-term traffic plume signals as "outliers." However, such short-term and abrupt fluctuations are key features of urban CO2 dynamics. Applying too long a window could remove meaningful short-term events as noise. Please provide additional justification or discussion on this issue.

**Response:**

85

We thank the reviewer for raising this important point and agree that different use cases of urban  $CO_2$  measurements require different approaches regarding temporal resolution and the treatment of local contamination. The chosen window size for the Hampel filter is intentional, as our objective is to extract well-mixed urban background signals while filtering out short-term plumes.

Our sensors are installed at rooftop level, typically surrounded by buildings with combustion-based point sources. While we try our best to select locations to minimize direct contamination, avoiding local influences entirely in dense urban environments is inherently difficult.

We would like to clarify that our implementation of the Hampel filter does not remove any data. Instead, it provides a flag indicating potential contamination events. We do this by comparing the output of the hampel filter with the original signal and flagging differences. This allows users with different scientific objectives, for instance studies focusing on short term plumes to apply their own filtering strategies using the published dataset. We have clarified this point in the revised manuscript.

**Addition to the manuscript:** (Page 11, After Line 246)**

In our implementation, no data points are removed. Potential contamination events are identified by comparing the Hampel filter output with the original signal and flagging deviations directly on the original time series. This approach allows users with different scientific objectives, for example those focusing on short-term plumes, to apply their own filtering or thresholding strategies using the published dataset.

(5) (Page 21, Figure 10, Lines 416-427) To control excessive local pollution, the study applied the Hampel filter used in previous studies. While this method effectively removes extremely high peaks, it does not fully eliminate local contamination. The paper notes that the filter captured the ventilation effects but did not perform particularly well. Moreover, since this station is used as a background site, placing the sensor so close to a ventilation outlet seems questionable. Please provide further explanation or justification for this site configuration.

**100 Response:**

105

We thank the reviewer for this constructive comment. At the beginning of the network deployment, limited guidance was available on what constitutes an optimal site configuration for urban  $\mathrm{CO}_2$  monitoring. Identifying rooftop hosts willing to provide space and electrical access free of charge proved challenging, and site selection therefore required some degree of compromise. We considered it valuable to include a diverse range of sites to better understand how varying local conditions influence measurement quality and concluded to deploy it rather than keeping it in our lab.

For the specific site mentioned, the air inlet was installed upstream of the prevailing wind direction relative to the nearby ventilation outlet. Under typical south-westerly winds and higher wind speeds, the inlet is expected to remain outside the plume, ensuring that background conditions are captured reliably. The potential contamination source operates only during the heating season and thus has limited temporal impact on the overall dataset.

We are happy to report that in September 2025 the administration of Maisach supported us to relocate the station to the building on the other side of the street, improving the site configuration.

---

## Author Comment (AC3)

**Response to Anonymous Referee #3 for EGUSPHERE-2025-4157 (ACROPOLIS: Munich Urban $CO_2$ Sensor Network)**

Patrick Aigner[1], Jia Chen[1], Felix Böhm[1], Mali Chariot[2], Lukas Emmenegger[3], Lars Frölich[1], Stuart Grange[3,4], Daniel Kühbacher[1], Klaus Kürzinger[1], Olivier Laurent[2], Moritz Makowski[1], Pascal Rubli[3], Adrian Schmitt[1], and Adrian Wenzel[1]

[1]Technical University of Munich (TUM), Munich, Germany
[2]Laboratoire des Science du Climat et de l'Environnement (LSCE/IPSL), Gif-sur-Yvette, France
[3]Swiss Federal Laboratories for Materials Science and Technology (Empa), Dübendorf, Switzerland
[4]University of Bern, Bern, Switzerland

**Correspondence:** Patrick Aigner (patrick.aigner@tum.de) and Jia Chen (jia.chen@tum.de)

**Anonymous Referee #3**

**General Comment**

*Summary. This manuscript presents the setup and testing of a network of 20 mid-cost CO2 concentration sensors with the aim to set up an urban network of CO2 concentration observations to identify urban gradients in CO2. The manuscript fits the scope*

5 *of the journal. The study is impressive and one of the first that illustrates the details in setting up, maintaining and interpreting such urban CO2 network. So I recommend to accept this manuscript after some revisions. Recommendation: Minor revisions are needed*

**Response:**

We sincerely thank the reviewer for the positive and detailed feedback, as well as for the time and effort dedicated to

10 reviewing our manuscript. We address the individual comments point by point below.

**Major Remarks**

(1) *The paper misses in my opinion the opportunity to explain why such a CO2 concentration network is needed. Most studies deal with CO2 fluxes, rather than concentrations in order to study the carbon budget of sources and sinks. Here you add a*

15 *network of CO2 concentration observations, but the study does not say much about how this complements or support the CO2 flux budget estimations, or how it can help to study CO2 advection estimates (since you generate spatial gradients that are unique).*

**Response:**

We thank the referee for this thoughtful comment. We agree that the manuscript should better explain the broader purpose

20 of an urban $CO_2$ concentration network. The primary focus of this paper is the development, deployment, and evaluation of the ACROPOLIS measurement network. Its role within ICOS Cities is to provide high-quality, spatially distributed $CO_2$

concentration data that can serve as a key observational input for urban inverse-modelling systems, in combination with detailed urban emission inventories.

The inverse modeling in ICOS Cities is developed by dedicated partner teams. Several results based on these efforts are already available (Brunner et al. (2025), Ponomarev et al. (2025)), while additional studies will be published soon. This division of work reflects the structure of the project, in which measurement, inventory development, and modelling are coordinated but addressed in separate contributions.

We have updated the Introduction and refined the Conclusion to clarify the intended role of ACROPOLIS within this framework and to emphasise that the present study focuses on establishing the measurement network and its data quality, while the inverse-modelling applications are addressed in companion work by colleagues.
* * *
**Addition to the manuscript:** (Introduction)

Urban $CO_2$ concentration networks such as ACROPOLIS provide observational constraints that are essential inputs to urban inverse-modelling systems when combined with detailed emission inventories (Lauvaux et al. (2020), Nalini et al. (2022)). These concentration measurements capture the integrated influence of local fluxes and atmospheric transport, enabling inverse models to infer spatial emission patterns and reduce uncertainties in city-scale carbon budgets. Within the ICOS Cities project, the modelling framework and the development of urban emission inventories are carried out by dedicated partner teams. Several results for Paris and Zurich are already available (Brunner et al. (2025), Ponomarev et al. (2025)), and additional studies will be published soon.
* * *
**Update of the manuscript:** (Conclusion)

Looking ahead, the ACROPOLIS network is expected to provide an important observational component within the broader ICOS Cities framework, and it may also support other projects making use of the open-source data products. A major upcoming step in the project is the integration of these $CO_2$ concentration observations into the urban modelling frameworks developed by partner teams. As these efforts advance, the network presented here may contribute to improved constraints on urban carbon budgets.
* * *
(2) *I think the manuscript can do more to justify the sensor network is more or less free from local influences. I fully agree with the strategy to find measurement locations like schools and hospitals and independent buildings to limit local influences, but at the same time the paper does not say/claim/justify one succeeded in doing so (which is not an easy task, I understand). This could be made more clear.*

**Response:**

We thank the referee for raising this important point. In dense urban environments, some degree of local contamination is unavoidable, and we now make this more explicit in the manuscript. Our strategy aims to minimise such influences by selecting the best available locations within the practical constraints of available spaces. Nevertheless, additional local effects can occur and must be addressed during post-processing.

To support this, all stations are equipped with co-located wind measurements, and our pipeline applies a Hampel-based spike detection scheme to identify local contamination. Across the network, the fraction of flagged 1-min values is generally low (median < 1 %), while stations with known nearby sources show higher fractions. We added a table summarising these outlier rates to make the prevalence of such events more transparent. Persistent or systematic contamination is further treated in the yearly release Level-2 datasets, where episodes attributable to identifiable local sources are manually flagged. We have updated the manuscript to clarify these limitations and to better describe how local influences need to be handled within the data workflow.
* * *
**Update of the manuscript:** (Section 2.3.4)

Although careful site selection reduces the influence of nearby emission sources, some level of local contamination is unavoidable in dense urban environments. The Hampel filtering therefore provides a first automated diagnostic of short-lived disturbances. Following this step, the Level 2 (L2) data set is produced through manual operator validation, during which persistent or recurrent anomalies attributable to identifiable local sources are flagged. The resulting one-minute and hourly averaged L2 data supplement the automated filtering L1 data. Both L1 and L2 datasets are uploaded to the ICOS Cities Portal, with L1 available continuously and L2 released once per year.
* * *
**Addition to the manuscript:** (New Section 2.5.2)

Some degree of local influence is unavoidable in dense urban environments, even with careful site selection. The availability of suitable installation locations is limited, and the deployed stations therefore represent the best feasible choices within these constraints. Remaining local effects are handled in the post-processing workflow described in Section 2.3.4, which includes automated Hampel filtering and manual validation in the Level 2 data products. The percentage of one-minute values flagged by the Hampel filter varies between sites and provides a first indication of short-lived local disturbances. An overview of the percentage of flagged data for each site can be found in Table A1 in Appendix H.
* * *
**Addition to the manuscript:** (Table A1, Appendix H)

| Station | BALR | MAIR | NPLR | HARR | BOGR | DLRR | RDIR | BLUT85 | TAUR | BLUT48 |
|---|---|---|---|---|---|---|---|---|---|---|
| **Spikes (%)** | 2.43 | 2.31 | 1.30 | 1.07 | 1.04 | 0.80 | 0.74 | 0.70 | 0.68 | 0.63 |
| **Station** | TUMR v1 | SENR | FELR | GROR | TUMR v2 | FINR | SCHR | SWMR | PASR | |
| **Spikes (%)** | 0.64 | 0.64 | 0.63 | 0.65 | 0.60 | 0.53 | 0.52 | 0.50 | 0.43 | |

**Minor Remarks**

(1) **(Ln 209)** *What about extreme high RH? Does the system still work at RH between 95 and 100% or in fog and rainy conditions? These are usually troublesome. Have data been removed, and if so, how many?*

**Response:**

We thank the referee for raising this important point. The outdoor enclosure is heated, which keeps the internal relative humidity well below critical levels even during periods of high ambient humidity. Across all sites, the internal humidity rarely exceeds 60 %, and values above this threshold are extremely uncommon. To illustrate this, we added a histogram of the internal

relative humidity to the Supplement (Fig. 1). No data were removed based on humidity.

[Figure]

**Figure 1.** Distribution of over 12 million relative in-flow humidity measurements across all urban ACROPOLIS sites in Munich for data from January 2024 until October 2025. The vertical dashed lines indicate the mean and median. Interquantile range (IQR), kernel density estimate (KDE), and 2.5% and 97.5% percentiles are shown.

(2) **(Appendix A)** *please swap the x axis and the y axis, since the Picarro is your reference, and should be thus on the x axis (independent variable) and the system is your test case (dependent variable).*

**Response:**

We thank the referee for this helpful suggestion. We have updated the figure in Appendix A so that the Picarro reference measurements are now placed on the x-axis and the system observations on the y-axis, following standard conventions for dependent and independent variables.

(3) **(Figure 6)** *in the x axis MAE and RMSE need a unit. The caption should be elaborated since it is not clear what is the meaning of a dot (i.e. is one dot one sensor?), and it is not clear how the RMSE and MAE are calculated. I.e. is it the RMSE over the all hourly values, daily values, daytime values... Some more guidance in the caption is welcome.*

**Response:**

We thank the referee for these helpful suggestions. Regarding the unit, we respectfully note that the y-axis already specifies ppm, which applies to both RMSE and MAE. To improve clarity, we updated the caption to explicitly describe how RMSE and MAE are computed and to clarify that each point corresponds to one ACROPOLIS system.

**Update of the manuscript:** (Figure 6, Caption)

Root mean square error (RMSE) and mean absolute error (MAE) of each GMP343 sensor compared to a Picarro G2301 reference instrument. For both RMSE and MAE, the plot contains 20 points corresponding to the 20 ACROPOLIS systems. All error metrics are calculated from hourly mean concentrations and reported in ppm.

(4) *I would like to see some more justification for the chosen error metric (e.g. based on*

80 *https://gmd.copernicus.org/articles/15/5481/2022/ and underlying works). RMSE and MAE are likely common practise, but RMSE is not an unbiased error estimator (https://www.sciencedirect.com/science/article/abs/pii/S0020025521011567 ), which may mean your results are better than you present now in the manuscript.*

**Response:**

We thank the reviewer for this constructive comment. Following Hodson (2022), RMSE and MAE should not be treated as

85 interchangeable metrics, as they reflect different assumptions about the underlying error distribution. RMSE is most suited for approximately Gaussian errors, whereas MAE provides a more robust estimate when the distribution shows heavier tails or a sharper central peak.

We calculated RMSE/MAE ratios following the recommendation in Karunasingha (2022) and added the results to Table 2. The average ratio across all sensors is 1.24, which is close to the theoretical value for Gaussian errors. Individual sensors span

90 values from 1.11 to 1.38, indicating that some sensors deviate from normality. Presenting both RMSE and MAE therefore gives a more complete description of sensor performance at the individual-sensor level.

To further characterise the error structure, we plotted the NDIR–Picarro residuals for the 2024 side-by-side comparison and for both systems in the 2025 evaluation (Fig. 2). These distributions confirm that specific sensors can exhibit Laplace-like residuals with a sharp peak and heavier tails, while in this example the Generation 2 system tends toward more Gaussian shape.

95 While RMSE and MAE are standard practice, RMSE remains the primary metric used throughout this study because it captures occasional larger deviations that can arise from environmental sensitivity and therefore reflects the upper bound of performance relevant for many applications. MAE is reported alongside RMSE to quantify the typical magnitude of errors and to characterise sensors with non-Gaussian residuals more robustly. Presenting both metrics improves transparency and allows direct comparison with studies that rely on either measure.

**Update of the manuscript:** (Table 2)

100 We added an addition column showing the RMSE/MAE ratio for each sensor to give an indication on the individual error distribution.

(5) **(Table 2, header)** *Mean bias, MAE and RMSE should have a unit*

**Response:**

We thank the referee for pointing this out. The table header has been updated to include the units for mean bias, MAE, and RMSE.

105

[Figure]

**Figure 2.** The figure shows the error distribution for all systems in the 2024 side-by-side campaign and for both system 3 (Generation 2) & system 6 (Generation 1) in the 2025 side-by-side campaign. We fit a normal distribution (blue line) and a Laplace distribution (red line) to the residuals (ACROPOLIS - Picarro) to help illustrate the different error distributions.

(6) **(Table 2)** *Reword Mean bias to bias, since bias is by definition an mean (as long as you do not average over all systems - which you do not do here).*

**Response:**

We thank the referee for this helpful suggestion. We have updated the table header by replacing "Mean bias" with "Bias".

110

(7) **(Figure 7)** *caption: reword "sensor temperature" to "hourly mean sensor temperature".*

**Response:**

We thank the referee for the helpful suggestion. We have updated the caption to state "hourly mean sensor temperature" for improved clarity.

115

(8) **(Figure 7)** *revise y axis label. The graph now suggests the temperature measurement is accurate at 0.01 K, which is not the case. In your wording in ln 371, you also use only 1 decimal.*

**Response:**

We thank the referee for this helpful remark. We adjusted the y-axis formatting to one decimal so that it matches the level of

120 precision discussed in the manuscript.

(9) **(Ln 377)** *significant improvement. Please add the results to a statistical test that confirms this statement.*

**Response:**

We thank the referee for pointing this out. We revised the wording to "clear improvement" to avoid implying statistical

125 significance.

> **Update of the manuscript:** (Line 395–399)
>
> Sensor 3 in the second generation system performed with an RMSE of 0.60 ppm, an MAE of 0.49 ppm, and a standard deviation of 0.52 ppm, indicating a clear improvement in precision compared to its performance in the first-generation system (RMSE 0.91 ppm, MAE 0.68 ppm, standard deviation 0.91 ppm).

(10) **(Figure 10, caption)** *Please add the height at which the wind speed was measured.*

**Response:**

We thank the referee for the helpful suggestion. We have updated the caption of Figure 10 to include the height at which the wind speed was measured (15 m AGL).

(11) **(Figure 11, caption)** *reword to "Time series of observed CO2 concentrations. . . . "*

**Response:**

We thank the referee for the suggestion and have updated the caption accordingly.

(12) **(Figure 12)** *profiles-> evolution. Profile is more reserved for vertical profiles.*

**Response:**

We thank the referee for the suggestion and have updated the caption accordingly.

(13) **(Ln 467-469)** *can be removed, since a) it is 3 short-sentences paragraph (too short), but mostly it reads as a figure caption, so should belong to the caption of fig 12 and not in the main text.*

**Response:**

We thank the referee for this helpful comment. We removed the three caption-like sentences from the main text and incorporated the relevant information into the end of Section 3.7.1.

> **Update of the manuscript:** (Line 497–499)
>
> GROR exhibits the lowest daytime concentrations across all seasons, reflecting its upwind position and correspondingly reduced exposure to urban emissions from Munich.

(14) **(Ln 471)** *This reflects enhanced photosynthetic uptake and higher boundary-layer heights. This statement is not confirmed with additional measurements. Are these available? I would say the concentrations are first of all lower because lower emissions in spring and summer than in winter. So a car traffic count or emission databases could support these.*

**Response:**

We thank the reviewer for this helpful comment. We agree that lower emissions in spring and summer are a primary driver of the observed seasonal decrease in $CO_2$ concentrations. We therefore revised the manuscript wording to clarify that our discussion of photosynthetic uptake and boundary-layer dynamics represents an interpretation rather than a conclusion supported

by additional measurements. At present, we do not include direct emission or traffic activity datasets in this analysis, but we acknowledge the reviewer's point and have adjusted the text to avoid implying observational confirmation.
* * *
**Update of the manuscript:** (Section 3.7.1)

We believe that the lower $CO_2$ levels during this period are primarily driven by reduced emissions, and that enhanced photosynthetic uptake together with higher boundary-layer heights further contribute to the observed seasonal decrease.
* * *
(15) **(Fig 13, caption)** *The results reveal distinct seasonal and spatial patterns in diurnal CO2 variability. This sentence should be removed, it is interpretation of the figure and thus needs to be in the main text.*

**Response:**

We thank the referee for pointing this out. We have removed the interpretative sentence from the caption as suggested.

(16) **(Fig 13, caption)** *"The diurnal variation is defined as the daily maximum minus minimum CO2 concentration." Do you mean "The diurnal variation is defined as the daily maximum minus minimum hourly CO2 concentration." ?*

**Response:**

We thank the referee for the clarification. Yes, the definition refers to hourly concentrations. We have updated the caption accordingly.
* * *
**Update of the manuscript:** (Figure 13, caption)

Diurnal variation is defined as the daily maximum minus the minimum of the hourly mean $CO_2$ concentration.
* * *
(17) **(Fig 13: y axis)** *the label says: Mean hourly diurnal cycle of CO2 variation. This is of course impossible (measuring an hourly diurnal cycle). I suggest to change to "Mean diurnal cycle of CO2 concentration (ppm) based on hourly mean observations"*

**Response:**

We thank the referee for the helpful remark. We have updated the y-axis label to "Mean diurnal $CO_2$ variation (max-min, ppm)", which accurately reflects the definition used (daily maximum minus minimum of the hourly mean concentrations) and avoids the ambiguity noted in the original wording.

(18) **(Figure 13)** *The figure's content is hyper-interesting and intriguing (and nicely plotted). But I was wondering whether an uncertainty estimate can be added to each (or a representative) label. E.g. for the urban station on the most left in the graph, the max diff between summer and winter is order 20 ppm. But if the error bar is 30 ppm (which I do not expect),then the differences between seasons are virtual. So if the error estimates are small, better to add them to show you have measured significantly different CO2 diurnal cycles between seasons. In fact you do in Fig 14!*

**Response:**

We sincerely thank the referee for the encouraging feedback on the figure and for raising this important point. As suggested, we now provide uncertainty estimates to clarify the robustness of the seasonal differences in diurnal $CO_2$ variation. Specifically, we added 95 % confidence intervals derived from bootstrap resampling of the hourly mean concentrations. These intervals demonstrate that the observed seasonal contrasts are well-resolved and not an artefact of sampling variability. The figure caption has been updated accordingly.
* * *
**Addition to the manuscript:** (Figure 13, caption)

Vertical bars indicate the 95 % confidence intervals derived from bootstrap resampling.
* * *
(19) **(Section 3.7.3)** *More justification is needed for the definition of the afternoon hours (12:00 - 18:00 local time). I do agree with your strategy to ignore nocturnal accumulation. However, in Munchen in mid winter the sunset is at 16:22 CET, which means you will have about 2 h of stratified atmosphere in your sample. Please explain why 10:00-16:00 local time, was not chosen as study period. Or whether that would have given other conclusions.*

**Response:**

We thank the referee for this thoughtful comment and agree that the afternoon period should account for the early sunset during mid-winter. We therefore adjusted the analysis window to 10:00–16:00 local time. This modification does not affect the overall interpretation or the seasonal contrasts discussed in this section. The manuscript has been revised accordingly.

**Additional Changes:**

- We updated Figures 12–14 with the most recent data (up to 20 November) and updated the manuscript accordingly. The overall patterns remain unchanged, although some values have shifted slightly due to the extended data period.

- We updated Appendix E to include available data covering all summer months. Patterns are as before, with more data to support the observed trends.

**References**

Brunner, D., Suter, I., Bernet, L., Constantin, L., Grange, S. K., Rubli, P., Li, J., Chen, J., Bigi, A., and Emmenegger, L.: Building-resolving simulations of anthropogenic and biospheric $CO_2$ in the city of Zurich with GRAMM/GRAL, Atmospheric Chemistry and Physics, 25, 14 387–14 410, https://doi.org/10.5194/acp-25-14387-2025, 2025.

210 Karunasingha, D. S. K.: Root mean square error or mean absolute error? Use their ratio as well, Information Sciences, 585, 609–629, https://doi.org/10.1016/j.ins.2021.11.036, 2022.

Lauvaux, T., Gurney, K. R., Miles, N. L., Davis, K. J., Richardson, S. J., Deng, A., Nathan, B. J., Oda, T., Wang, J. A., Hutyra, L., and Turnbull, J.: Policy-Relevant Assessment of Urban CO2 Emissions, Environmental Science & Technology, 54, 10 237–10 245, https://doi.org/10.1021/acs.est.0c00343, pMID: 32806908, 2020.

215 Nalini, K., Lauvaux, T., Abdallah, C., Lian, J., Ciais, P., Utard, H., Laurent, O., and Ramonet, M.: High-Resolution Lagrangian Inverse Modeling of CO 2 Emissions Over the Paris Region During the First 2020 Lockdown Period, Journal of Geophysical Research: Atmospheres, 127, https://doi.org/10.1029/2021JD036032, 2022.

Ponomarev, N., Steiner, M., Koene, E., Rubli, P., Grange, S., Constantin, L., Ramonet, M., David, L., Emmenegger, L., and Brunner, D.: Estimation of $CO_2$ fluxes in the cities of Zurich and Paris using the ICON-ART CTDAS inverse modelling framework, EGUsphere, 2025,
220 1–35, https://doi.org/10.5194/egusphere-2025-3668, 2025.